

# Rainfall disaggregation for hydrological modeling: Is there a need for spatial consistence?

Hannes Müller[1], Markus Wallner[2], Kristian Förster[1,3,4]

[1]Leibniz Universität Hannover, Institute of Hydrology and Water Resources Management, Leibniz Universität Hannover, Hanover, 30175, Germany
[2]bpi Hannover - Beratende Ingenieure, Hanover, 30177, Germany
[3]Institute of Geography, University of Innsbruck, Innsbruck, 6020, Austria
[4]alpS - Centre for Climate Change Adaptation, Innsbruck, 6020, Austria

*Correspondence to*: Hannes Müller (mueller@iww.uni-hannover.de)

**Abstract.**

In this investigation, the influence of disaggregated rainfall data sets with different degrees of spatial consistence on rainfall runoff modeling results is analyzed for three meso-scale catchments in Lower Saxony, Germany. For the disaggregation of daily rainfall time series into hourly values a multiplicative random cascade model is applied. The disaggregation is applied on a per station basis without consideration of surrounding stations, hence subsequent steps are then required to implement spatial consistence. Spatial consistence is here represented by three bivariate spatial rainfall characteristics, complementing each other. A resampling algorithm and a parallelization approach are evaluated against the disaggregated time series without any subsequent steps. With respect to rainfall, clear differences between these three approaches can be identified regarding bivariate spatial rainfall characteristics, areal rainfall intensities and extreme values. The resampled time series lead to the best agreement with the observed ones. Using these different rainfall data sets as input to hydrological modeling, we hypothesize that derived runoff statistics are subject to similar differences as well. However, an impact on the runoff statistics summer and winter peak flows, monthly average discharge and flow duration curve of the simulated runoff time series cannot be detected. Several modifications of the investigation using rainfall runoff models with and without parameter calibration or using different rain gauge densities lead to similar results in runoff statistics. Only if the spatially highly resolved rainfall-runoff WaSiM-model is applied instead of the semi-distributed HBV-IWW-model, slight differences regarding the seasonal peak flows can be identified. Hence, the hypothesis formulated before is rejected in this case study. These findings suggest that (i) simple model structures might compensate for deficiencies in spatial representativeness through parameterization and (ii) highly resolved hydrological models benefit from improved spatial modeling of rainfall.



# 1. Introduction

Flood quantiles are important information for the creation of flood hazard maps, the construction of riverfront buildings and landscape development plans, for example. For ungauged catchments and catchments with short discharge observation periods, rainfall-runoff modeling is a possibility to obtain long, simulated discharge time series which can then be used for

derived flood frequency analysis.

The most important data input for rainfall-runoff modeling are rainfall time series (Beven, 2001). Melsen et al. (2015) gave an overview of typical processes for different catchment sizes and corresponding temporal resolutions. For catchments with areas of a few hundred square kilometers, time series with hourly resolutions are required for the simulation of instantaneous flood peaks. In most of these cases, observed rainfall time series of that kind are too short or the network density is too low.

This is an issue, because it limits the length of the simulation period and hence the derivable flood frequencies.

Usually, time series of non-recording stations have much longer observation periods and a higher network density. Using information from the time series of recording stations, time series from non-recording stations can then be disaggregated. One possible method for the disaggregation of rainfall is the multiplicative random cascade model (e.g. Olsson, 1998), which was originally introduced within the field of turbulence theory (Mandelbrot, 1974). The use of observed daily time series as

input is a strong advantage of the cascade model over other rainfall generators (e.g. Poisson-cluster models (Rodriguez-Iturbe et al., 1987, Onof et al., 2000)), since starting with "true" rainfall amounts and intermittency facilitates their conservation to higher temporal resolutions.

With the micro-canonical cascade model, the rainfall amount of a coarse time step (e.g. a day) is conserved exactly through the disaggregation process, so that an aggregation of the disaggregated time series would result exactly to the original

observed time series. Starting from a daily resolution, an hourly temporal resolution is achieved, which is a convenient input resolution for many rainfall-runoff models. However, this disaggregation method is a univariate process, carried out for single time series only which are independent from the time series of surrounding stations. Through the systematically random distribution of the rainfall amount within a day, unrealistic patterns of rainfall are generated and the spatial consistence of rainfall is missing. If an unrealistic spatial distribution of rainfall is used within a rainfall-runoff simulation, it

can be assumed that this affects the simulated runoff. Müller and Haberlandt (2015) have introduced a resampling scheme as a subsequent step after the disaggregation process, which can be used for the implementation of spatial consistence within disaggregated time series. Spatial consistence is hereby defined by three bivariate rainfall characteristics: the probability of occurrence, the coefficient of correlation after Pearson, and the continuity ratio (Wilks, 1998). The implementation of spatial consistence for hourly time series was proven by the above mentioned bivariate characteristics in addition to areal rainfall

intensities resulting from the disaggregated time series. Without resampling, areal rainfall intensities were underestimated. The resampling algorithm was additionally tested for time series of 5 min resolution by Müller and Haberlandt (2016). Bivariate rainfall characteristics as well as the simulated runoff from an artificial sewage system were positively validated against observed rainfall time series and its resulting simulated runoff.



Haberlandt and Radtke (2014) overcame the lack of spatial consistence using a parallelization approach. For each day they chose the relative diurnal cycle from the disaggregation time series of the station with the highest daily rainfall amount and then adopted this diurnal pattern for all other time series. This procedure leads to simultaneous rainfall events for all stations with registered rainfall for that day. This homogenous rainfall could lead to an overestimation of simulated floods, but is

acceptable in comparison to a possible underestimation by the before mentioned unrealistic spatial rainfall patterns. However, Ding et al. (2016) also used disaggregated time series for their rainfall-runoff analyzes with a focus on instantaneous peak flows, but without any subsequent changes to the disaggregated time series. Neither a systematic over- or underestimation of simulated discharge and flood peaks can be found in both investigations.

It can be questioned, why the simulation results from both investigations, based both upon unrealistic spatial rainfall

behavior, leads to an acceptable representation of observed discharge characteristics. The hypothesis of this study is that rainfall data sets with different degrees of spatial consistence will result in different areal rainfall intensities and hence influence runoff statistics derived from simulated runoff time series. Therefore, three different rainfall data sets are used as input for rainfall runoff modelling: disaggregated time series with (Müller and Haberlandt, 2015) and without (Ding et al., 2016) implemented spatial consistence, and thirdly time series with an "overestimated spatial consistence" by parallelization

(Haberlandt and Radtke, 2014). A systematic comparison is carried out including rainfall-runoff simulations with and without calibration, differing station densities and different rainfall-runoff models.

In general, calibration and validation of rainfall-runoff model parameters are carried out by a quantitative comparison of simulated and observed time series. This strategy is not applicable by using disaggregated rainfall time series as input, since the daily rainfall amount is distributed randomly in time during a day. Hence, the temporal connection between rainfall and

runoff is missing. An alternative strategy is the calibration on runoff statistics and has been applied before by amongst others Yu and Yang (2000), Westerberg et al. (2011), Haberlandt and Radtke (2014), Wallner and Haberlandt (2015) and Ding et al. (2015). Runoff statistics have no connection to time, but contain useful information about the hydrograph and hence about the hydrological regime and its characteristics. It is assumed, that by a simultaneous consideration of different complimentary runoff statistics, the runoff behavior can be represented sufficiently. Possible runoff statistics are: runoff

extremes for different seasons of a year (to take into account different genesis), flow duration curves (to describe the overall behavior), and average monthly values (to describe the inter-annual variability).

The manuscript is organized as follows: after a brief description of the investigation area and the data in chapter 2, the rainfall generation including the implementation of spatial consistence and the applied rainfall-runoff models including the calibration technique are explained in chapter 3. Chapter 4 includes the results for both the rainfall generation and rainfall-

runoff-modeling. A summary of the rainfall-runoff model results is provided in chapter 5 and general conclusions and a brief outlook are provided in chapter 6.



## 2. Data & Investigation area

### 2.1 Catchments

The investigation is carried out for three catchments in the Aller-Leine river basin, namely Reckershausen, Pionierbrücke
and Tetendorf (see Fig. 1). The river basin is situated in Lower Saxony, Northern Germany. Based on the Köppen-Geiger

5    climate classification, the river basin can be divided into a temperate oceanic climate in the north and a temperate continental
climate in the south (Peel et al. 2007). For Reckershausen an additional investigation regarding rain gauge network density is
carried out. All recording and non-recording stations for Reckershausen are shown in Fig. 2.

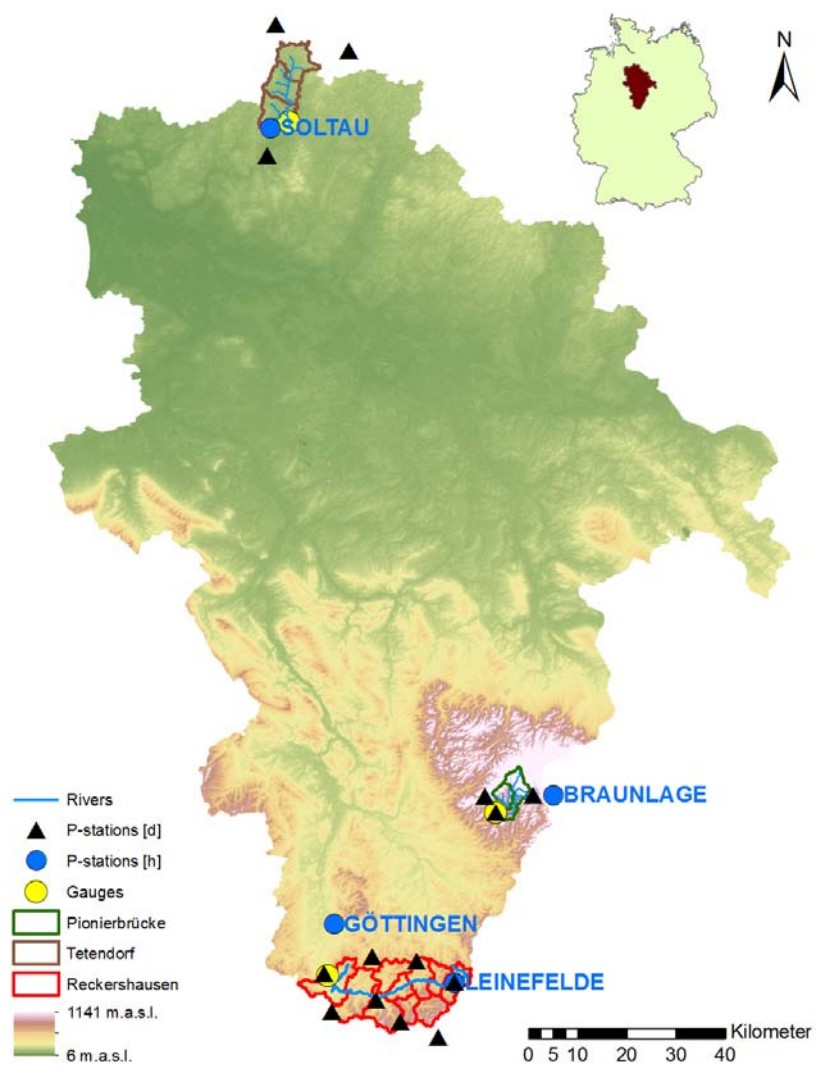

10    **Fig. 1. Location of all three catchments in the Aller-Leine-river basin and its location in Germany.**


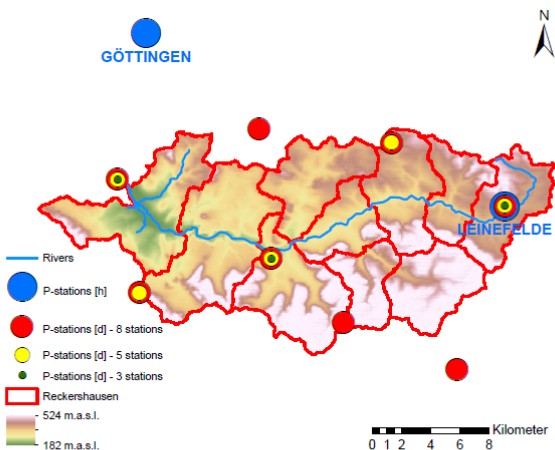

**Fig. 2. Catchment Reckershausen including sets of 3, 5 and 8 non-recording stations used for network density analysis.**

The catchments differ concerning area and elevation as well as land use and soil conditions. A brief description can be found in Table 1. The soil information is extracted from the soil map BÜK1000 of the Federal Republic of Germany with a scale of 1:1,000,000 (Hartwich et al., 1998). Information regarding the land use is extracted from the CORINE database (Federal Environment Agency, 2009). The time of concentration has been estimated as per Kirpich (1940).

**Table 1. Brief description of the investigated catchments with fraction of dominant soil type and land use**

| Catchment | River | Area [km²] | Sub-catchments | Time of concentration [h] | Dominant soil type | Dominant land use |
|---|---|---|---|---|---|---|
| Pionierbrücke | Sieber | 44 | 2 | 1.8 | Spodic Cambisols (77 %) | Coniferous forest (81 %) |
| Tetendorf | Böhme | 110 | 3 | 7.2 | Haplic Podzols / Dystric Regosols (40 %) | Non-irrigated arable land (39 %) |
| Reckers-hausen | Leine | 321 | 10 | 7.4 | Dystric Cambisols (37 %) | Non-irrigated arable land (59 %) |

## 2.2 Climate data

For the rainfall disaggregation, time series of recording and non-recording stations are required. Time series of the recording stations are used for the parameter estimation of the cascade model (described in chapter 3.1 a), which is in turn used for the disaggregation of the time series of the non-recording stations. An overview of rain gauges and measuring periods used in this study is given in Fig. 1 and Table 2. For the non-recording stations, the chosen period is the longest available period

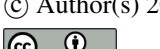



with data for all stations in a catchment. From Table 2 it can be seen, that time series lengths have a longer duration for non-recording stations in comparison to those from recording stations for all catchments (up to 2.7 times for Pionierbrücke). Additionally, the number of non-recording-stations is higher.

**Table 2. Rain gauges and time series lengths used for each catchment**

| Catchment | Rain gauges | Type | Start | End |
|---|---|---|---|---|
| Pionierbrücke | Rehberger Grabenhaus Sieber Hauskuehnenburg | Non-recording | 1950 | 2004 |
| | Braunlage | Recording | 1993 | 2013 |
| Tetendorf | Bispingen-Huetzel Fallingbostel Schneverdingen | Non-recording | 1984 | 2006 |
| | Soltau | Recording | 1993 | 2013 |
| Reckershausen | Wachstedt Leinefelde Heiligenstadt-Kalteneber Reinholterode Uder Bornhagen Friedland (Lower Saxony) Gleichen-Elbickerode | Non-recording | 1972 | 2006 |
| | Göttingen Leinefelde | Recording | 1993 | 2013 |

5    For the rainfall-runoff-model HBV (see section 3.2), time series of precipitation, temperature and potential evaporation are needed. The following description of data processing of temperature and potential evaporation is based on Wallner et al. (2013) and was carried out for the whole Aller-Leine basin. The temperature time series were derived through an interpolation using External Drift Kriging of 38 recording stations with hourly resolution, whereby the additional information is elevation.

10   The calculation of the potential evaporation is carried out using the Turc-Wendling method on a daily basis (DVWK, 1996). The required sunshine duration per day was derived through Ordinary Kriging using 29 stations . To achieve an hourly resolution, daily values have been divided by 24, since the inter-daily distribution of potential evaporation has been shown not to be that sensitive as model input. Different land use types have been taken into account by using an average land use




parameter (DVWK, 2002) similar to the crop coefficient. All input data were interpolated and subsequently aggregated to subcatchment scale.

For the WaSiM model, which is applied only for the catchment Pionierbrücke, climate time series are needed as point or gridded information on an hourly basis. From the climate station Braunlage, time series of temperature, relative air humidity, and wind speed are available with an hourly resolution. Global radiation was only available on a daily basis, but has been disaggregated to hourly values using an approach as per Förster et al. (2016).

## 2.3 Runoff data

The available discharge data of the three catchments is listed in Table 3. While observed time series are only available since 2000 (Pionierbrücke) and 2004 (Tetendorf and Reckershausen), observed extreme values exist for much longer periods. Daily discharge time series exist for at least as long as the period of the monthly extreme values.

For the calibration, a special focus is given to the extreme values of the summer (01.05.-31.10.) and winter period (01.11.-30.04.). Therefore, the maximum observed value of each half year were extracted from both data sources to generate periods as long as possible.

**Table 3. Available periods of runoff data types**

| Catchment | Hourly discharge time series | Daily discharge time series | Monthly extreme values |
|---|---|---|---|
| Pionierbrücke | 2000-2013 | 1929-2006 | 1952-2005 |
| Tetendorf | 2004-2013 | 1986-2000 | 1986-2000 |
| Reckershausen | 2004-2009 | 1964-2006 | 1974-2005 |

## 3. Methods

The method chapter consists of two subchapters. In the subchapter 3.1, the multiplicative cascade model for the disaggregation of rainfall time series is explained. Additionally, two methods for the implementation of spatial consistence in the disaggregated time series are presented. The descriptions of the two rainfall-runoff models HBV and WaSim and the calibration procedure for HBV can be found in subchapter 3.2.

## 3.1 Rainfall generation

### a) Rainfall disaggregation

The most important input for rainfall-runoff models are long and high-resolution rainfall time series from a dense rain gauge network. Data of that kind is not available in most cases. However, time series from non-recording stations (e.g. daily




resolution) are often available for long periods and with a high spatial density. The objective of the disaggregation is to generate high-resolution time series from the time series of the non-recording stations with information from the observed, high-resolution data.

The multiplicative random cascade model (Müller and Haberlandt, 2015) is applied for the disaggregation. A general scheme

of this model is shown in Fig. 3. One coarse time step is divided into *b* finer time steps of equal length. The branching number b determines the number of finer time steps and is in the first disaggregation time step *b=3* and in all following disaggregation steps down to 1 h resolution *b=2*. The cascade-model is micro-canonical, so the rainfall amount of each time step is conserved exactly. A re-aggregation of the disaggregated time series yield the observed time series used for the disaggregation. Since the focus of this investigation is not on the disaggregation itself, only a brief description of the model

is included below. For a more detailed explanation, the interested reader is referred to Müller and Haberlandt (2015).

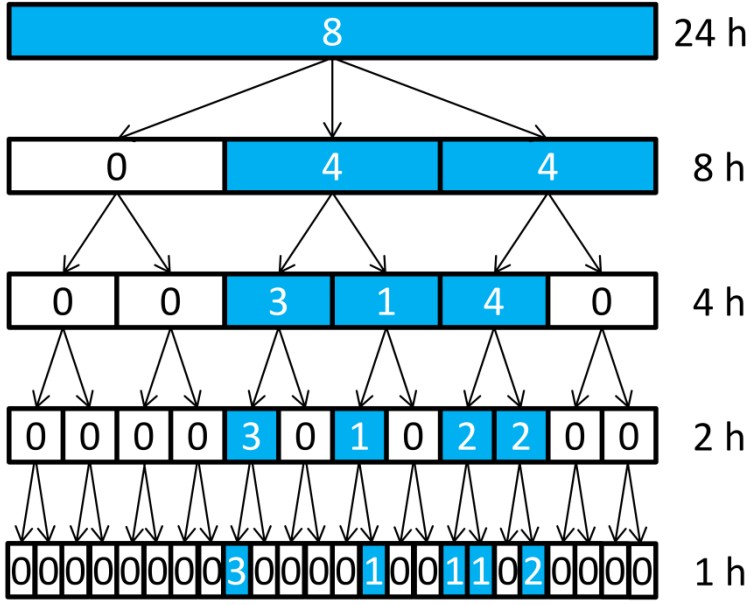

**Fig. 3. General disaggregation scheme of the applied multiplicative cascade model (values inside the boxes represent rainfall amount, blue or white box color indicates wet or dry time steps, respectively)**

All parameters can be extracted directly from the observed, high-resolution time series by a reverse application of the cascade model (Carsteanu and Foufoula-Georgiou, 1996). For the parameter estimation, the position and volume class of the wet time step to disaggregate is taken into account. The position describes the wetness state of the preceding and succeeding time steps, so that four states can be distinguished: starting, enclosed, ending, and isolated position. For each position class, two volume classes are distinguished, whereby the arithmetic mean of all rainfall intensities for each position is chosen as

the threshold between the lower and upper volume class. The structure of different position and volume classes in the observed time series and its conservation during the disaggregation process was found before by e.g. Buishand (1977),



Olsson (1998), Rupp et al. (2009) or Thober et al. (2014). Only for the first disaggregation step (from 24 h to 8 h) is no position taken into account (Müller and Haberlandt, 2015). While a uniform distribution is used for the first disaggregation step for the splitting of the rainfall amount on the number of as wet identified time steps, for *b=2* empirical distribution functions are used. For the sake of completeness it should be mentioned here that an unbounded cascade model has been used (Marshak et al, 1994). For all disaggregation steps with *b=2*, the same parameter set is applied due to the findings of scale invariance over these temporal resolutions (see Veneziano et al. (2006) and references therein).

### b) Bivariate characteristics

The disaggregation of single time series is carried out without taking into account time series of surrounding stations. For each time series the cascade model distributes the wet time steps randomly during a wet day due to its disaggregation scheme. Hence, spatial consistence of rainfall is underestimated after the disaggregation. Spatial consistence is defined in this investigation by bivariate spatial rainfall characteristics, the namely probability of occurrence, Pearson's coefficient of correlation, and the continuity ratio (Wilks, 1998). These characteristics have been used before by e.g. Haberlandt (2008) and will be briefly described here:

The probability of occurrence $P_{k,l}$ describes the probability of rainfall occurrence at the same time at two stations *k* and *l*:

$$P_{k,l}\left(z_k > 0 \mid z_l > 0\right) \approx \frac{n_{11}}{n} \quad , \tag{1}$$

where *n* is the total number of non-missing observation hours at both stations, and the number of simultaneous rainfall occurrence at both stations is represented by $n_{11}$.

The Pearson's coefficient of correlation ρ describes the relationship between simultaneously occurring rainfall at two stations *k* and *l* as a measure of the linear relation between both rainfall time series (Eq. (2)). Breinl et al. (2014) used this coefficient before for multisite rainfall generation:

$$\rho_{k,l} = \frac{cov(z_k, z_l)}{\sqrt{var(z_k) \times var(z_l)}}, \; z_k > 0, \; z_l > 0 \qquad . \tag{2}$$

Müller and Haberlandt (2015) found an intensity-dependency for Pearson's coefficient of correlation and distinguished between ρ(k≤4 mm) and ρ(k>4 mm), which is adopted here.

The continuity ratio $C_{k,l}$ compares the expected rainfall amount at one station for times with and without rain at the neighboring station (*E.* is the expectation operator):

$$C_{k,l} = \frac{E(z_k|z_k>0, z_l=0)}{E(z_k|z_k>0, z_l>0)} \tag{3}$$

These characteristics are distance-dependent and prescribed values can be estimated as functions of the separation distance between two stations from observed data (see Fig. 4).



### c) Implementation of spatial consistence

As mentioned before, the disaggregation of single time series is a point-process with no surrounding stations taken into account. Input data sets for the rainfall-runoff models consisting of just the disaggregated time series without subsequent steps to implement spatial consistence are referred to as V1 (no implementation of spatial consistence). Two methods for the

implementation of spatial consistence and resulting in the rainfall data sets V2 and V3, are applied in this investigation.

The first method, resulting in V2, is based on simulated annealing (Aarts and Korst, 1965, Kirkpatrick et al., 1983), a non-linear optimization method from the group of resampling algorithms. The aim of simulated annealing is to modify the disaggregated time series and in doing so minimizing the objective function (Eq. 4) without changing the structure or the absolute daily totals of rainfall amounts, since exact conservation is an advantage of the micro-canonical cascade model. The

elements, which are swapped during the resampling process, are relative diurnal cycles. This enables the exact conservation of rainfall amount for each day. Swapping is only allowed between the same position and volume class (see chapter 3.1), which enables the structural conservation of the generated time series.

The following description of the resampling algorithm is based on Müller and Haberlandt (2015). Only a brief explanation is given here and the interested reader is referred to their investigation for further details.

One time series is chosen randomly from all disaggregated time series and used as the reference time series *R1* in the resampling process. A second time series *M1* is chosen randomly from the leftover disaggregated time series and will be modified during the resampling process. Relative diurnal cycles of *M1* are swapped with the aim to minimize the objective function:

$$O_{k,l} = w_1 \times \left( P_{k,l} - P^*_{k,l} \right) + w_2 \times \left( \rho_{k,l,\leq 4} - \rho^*_{k,l,\leq 4} \right) + w_3 \times \left( \rho_{k,l,>4} - \rho^*_{k,l,>4} \right) + w_4 \times \left( C_{k,l} - C^*_{k,l} \right) \qquad (4),$$

that summarizes the differences between prescribed values (indicated by *) and the actual values. The simulated annealing algorithm has the potential to leave local optima of the objective function by also accepting "bad swaps" (worsening of the objective function). Bad swaps are accepted with a certain probability based on the annealing temperature $T_a$, which is reduced during the resampling process. The fraction of bad swaps acceptance is decreasing during the resampling process. If the objective function does not improve after a certain number of iterations, the time series *M1* becomes an additional

reference time series *R2*. In the next run, a new randomly chosen time series *M2* from the leftover disaggregated time series will be fitted to the two reference time series *R1* and *R2*. For the minimization of the objective function, the average value of $O_{k,l}$ of all station pairs is used. This procedure repeats, as long as there are leftover disaggregated, non-modified time series.

The second method, resulting in rainfall data set V3, is a more pragmatic solution and was introduced by Haberlandt and Radtke (2014). For each day, the station with the highest rainfall amount is identified. The relative diurnal cycle of this

station is transferred to all other stations for this day. This parallelization is carried out for all days of the disaggregated time series. The varying diurnal distributions of rainfall at each station without spatial patterns, leading to an underestimation of spatial consistence, is instead transformed to a simultaneous occurrence of rainfall at all stations with an overestimation of spatial consistence.

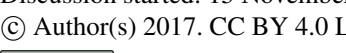


Both methods are compared against using the disaggregated time series without any subsequent steps. For analyzes and discussion of the impacts of these methods, the designations listed in the summarizing Table 4 are used.

**Table 4. Short characterization of the three rainfall data sets**

| Starting point | Subsequent step | Rainfall occurence at different stations | Designation |
|---|---|---|---|
| Disaggregated time series | none | Random | V1 |
| | Resampling | Intersecting | V2 |
| | Parallelization | Simultaneous | V3 |

## 3.2 Hydrological Models

For analyzing the impact of rainfall data sets with different spatial consistencies, two models, HBV-IWW (Wallner et al., 2013) and WaSiM (Schulla, 1997, 2015), are used. All simulations are carried out continuously. This enables the derivation of flood frequency analyzes and avoids uncertainties from unknown initial conditions resulting from event-based modeling (Pathiraja et al., 2016). Additionally, an initial phase of one year is used as a spin-up period to achieve plausible initial conditions for all storages.

### a) HBV-IWW including calibration procedure

The HBV-IWW model is based on the HBV model that was originally developed at the Swedish Meteorological and Hydrological Institute (SMHI) in early 1970s (SMHI, 2008) and was modified by Wallner et al. (2013). HBV-IWW, for simplification titled HBV, is a conceptual model, where runoff generation and runoff transformation are represented by simple relationships between storage and effective precipitation, respectively runoff (see Fig. 4). Snow accumulation and melt is based by a threshold temperature and the degree day method. After the snow storage, all precipitation and snow melt is entering the soil storage where actual evaporation is considered. Depending on the state of the soil storage, water is released to the upper groundwater layer from where surface runoff and interflow can occur. Both are controlled by a storage coefficient. Water from the upper groundwater layer can also percolate to the lower groundwater layer. The outflow from the latter is representing the baseflow component. Surface runoff, interflow and baseflow are finally summarized and transformed via a triangular unit hydrograph. River routing is carried out via the Muskingum method. Further details about the model parameters (Table 5) can be found in Wallner et al. (2013).

The calibration of rainfall-runoff models is traditionally carried out by modifying the model parameters so that the simulated hydrograph best represents the observed hydrograph (e.g. Beven, 2001). However this is not possible for disaggregated time series, since the rainfall is distributed randomly inside a day. For example, an observed peak flow can be reproduced perfectly in its magnitude in the simulated runoff time series, but will occur at a different time point. Hence, only runoff statistics can be used for the calibration. For this investigation, quantiles of the distribution functions fitted to the extreme





values of summer (Extr-Su) and winter (Extr-Wi), quantiles of the flow duration curve (FDC), and monthly averages (Q-mon) are used for calibration. The calibration procedure is illustrated in Fig. 4.

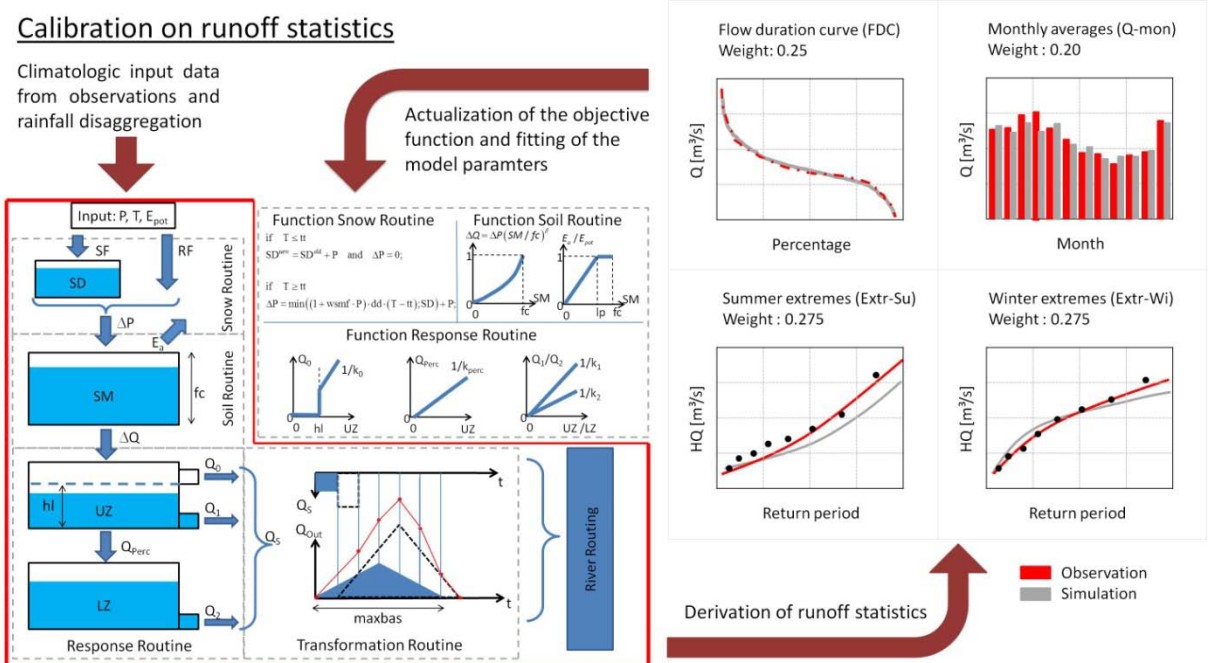

**Fig. 4. Flow chart and applied calibration procedure for HBV (Wallner and Haberlandt, 2015)**

For Extr-Su and Extr-Wi, a two-parametric Gumbel-distribution is fitted to the annual series of extreme values. L-moments are used for parameter estimation to reduce the sensitivity against outliers (Hosking and Wallis, 1997). Since extreme values represent only a small fraction of the discharge time series, FDC and Q-mon of the simulated discharge time series are used to represent the more frequent discharge values. Q-mon accounts for the temporal dependency on the inter-annual variation

of the discharge. FDC and Q-mon are calculated from averaged daily discharge values in order to reduce computation time. For the goodness-of-fit analyzes of simulated (Sim) and observed (Obs) statistics, the Nash-Sutcliffe-efficiency NSE (Nash and Sutcliffe, 1970) is used. A perfect fit would result in *NSE=1*, while assuming the average of the observed data for all time steps would result in *NSE=0*. The equation for the *NSE* is given in Eq. 5 and the corresponding quantiles for Extr-Su, Extr-Wi and FDC and months for the Q-mon, respectively, are given in Eq. 6.

$$NSE = 1 - \frac{\sum_{t=1}^{n}(Q_{Obs}(t) - Q_{Sim}(t))^2}{\sum_{t=1}^{n}(Q_{Obs}(t) - \overline{Q_{Obs}})^2} \rightarrow max \tag{5}$$

$$t = \begin{cases} \{0.05, 0.25, 0.5, 0.75, 0.95, 0.975\} & for\ FDC \\ \{1, 2, 3, 4, 5, 6, 7, 8, 9, 10, 11, 12\} & for\ Q - mon \\ \{0.2, 0.5, 0.8, 0.9, 0.95, 0.98, 0.99\} & for\ Extr - Su\ and\ Extr - Wi \end{cases} \tag{6}$$

The goodness-of-fit of all runoff statistics are summarized in the objective function $O_{stat}$, used for the calibration:

$$O_{stat} = 1 - (0.275 \cdot NSE_{Extr-Su} + 0.275 \cdot NSE_{Extr-Wi} + 0.2 \cdot NSE_{FDC} + 0.25 \cdot NSE_{Q-mon}) \rightarrow min \tag{7}$$





For the optimization simulated annealing is used. The parameters modified during the optimization with the corresponding ranges are given in Table 5. The periods for calibration and validation are listed in Table 6 for each catchment.

**Table 5. HBV model parameters modified during calibration with limiting ranges**

| Parameter | Unit | Explanation | Minimum | Maximum |
|---|---|---|---|---|
| $wsmf$ | [mm$^{-1}$] | Wet snow melt factor | 1 | 4 |
| $tt$ | [°C] | Threshold temperature | -1.5 | 1.5 |
| $dd$ | [mm°C$^{-1}$d$^{-1}$] | Degree day factor | 0.5 | 5 |
| $fc$ | [mm] | Field capacity | 50 | 300 |
| $lp$ | [-] | Limit for potential evapotranspiration | 0.1 | 0.95 |
| $\beta$ | [-] | Empirical factor for runoff calculation from the soil layer | 0.5 | 4 |
| $hl$ | [mm] | Threshold value for surface runoff | 1 | 30 |
| $k0$ | [d] | Storage coefficient surface runoff | 0.25 | 5 |
| $k1$ | [d] | Storage coefficient interflow | 3 | 40 |
| $k2$ | [d] | Storage coefficient baseflow | 50 | 500 |
| $kperc$ | [d] | Storage coefficient perculation | 3 | 40 |
| $maxbas$ | [h] | Length of the triangular unit hydrograph impulse | 3 | 10 |
| $mx$ | [-] | Weighting factor of Muskingum method | 0.1 | 0.4 |
| $mk$ | [h] | Retention constant of Muskingum method | 0.25 | 10 |

5 **Table 6. Calibration and validation period for all catchments**

| Gauge | Calibration period | | Validation period | |
|---|---|---|---|---|
| | Start | End | Start | End |
| Pionierbrücke | 01.11.1952 | 31.10.1977 | 01.11.1977 | 31.10.2003 |
| Tetendorf | 01.11.1986 | 31.10.1993 | 01.11.1993 | 31.10.2000 |
| Reckershausen | 01.11.1974 | 31.10.1990 | 01.11.1990 | 31.10.2006 |

**b) WaSiM**

WaSiM (Schulla, 1997, 2015) is a physically based and distributed hydrological model which has been designed to study climate change and land-use change impacts on the water balance and floods in meso-scale catchments (e.g., Niehoff et al. 10 2002, Bormann and Elfert, 2010). WaSiM was formerly known as WaSiM-ETH, but has since been renamed (Schulla, 2015) and hence the new abbreviation is used throughout the manuscript. WaSiM is flexible regarding the resolution of spatial

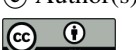



input data. In general, elevation, land-use, and soil data need to be prepared as gridded raster datasets. The spatial resolution of WaSiM applications covers several scales ranging from tens of meters to a few kilometers.

A set of alternative hydrological process representations for each of the following sub-models is included in the model in order to cover different user needs and meteorological data requirements: (i) evapotranspiration, (ii) snow, (iii) interception, and (iv) soil water. This list is not exhaustive since other processes can be also addressed using the model. Here, only the processes utilized in this study are described. Potential evapotranspiration is computed using the Penman-Monteith approach (e.g., Monteith, 1965) taking into account look-up tables of parameters defined for different land-use classes. Seasonal snow cover dynamics is simulated using a temperature threshold for phase partitioning and a temperature index model for snowmelt calculations. A bucket approach is applied to consider interception of rainwater. The soil water dynamics including actual evapotranspiration, infiltration, lateral outflow (interflow), and percolation is simulated in a numerical scheme which is based on the Richards equation. The lowermost nodes in each grid cell which are subject to saturation represent the groundwater storage in the model. A linear storage approach is applied here to simulate the outflow from the groundwater.

Since WaSiM is more complex than HBV with respect to computational needs, a different strategy for model calibration was chosen. As the number of both adjustable parameters and iterations is limited due to limited computational resources, a lexicographical approach was set-up for model calibration (Gelleszun et al., 2015). In this way, the optimization of parameters is divided into subsequent steps that are associated to different processes. In a first step, the parameters of the soil water balance and runoff generation (i.e. recession of hydraulic conductivity along the soil profile and the flow density) have been calibrated through maximizing *NSE*. Then, the baseflow recession is improved through minimizing the root mean square error *RMSE* of the lowermost part of the flow duration curve (two parameters). Both calibration steps have been performed using hourly meteorological time series and observed discharge time series from the period 2009-2012. As highly resolved meteorological observations are only available from 2000 onwards, an additional calibration step has been carried out using disaggregated rainfall time series in order to better match the long-term water balance characteristics through slightly modifying canopy resistance parameters of the evapotranspiration model. Without these pre-calibration steps an underestimation of the mean discharge and hence the water balance was identified. An incorrect representation of the water balance introduces other uncertainty sources which hence superpose the effects of the different versions of spatial rainfall. However, these pre-calibration was focused only on the water balance itself and not on the objectives used in Eq. (7).

## 4. Results & Discussion

For the discussion of the results, the chapter is divided into two parts. The first part deals with the interpretation of the rainfall spatial variability, while the influence on simulated discharges is discussed in the second part.



## 4.1 Rainfall

For the disaggregation of daily rainfall time series to hourly values, the micro-canonical cascade model of Müller and Haberlandt (2015) is used. This model was previously validated in the beforementioned study for the the Aller-Leine river basin, which is also considered in this study. Since the focus of this investigation is the spatial variability of the generated rainfall, the interested reader is referred to their investigation for a detailed analyzes of point results. In their study, observed high-resolution time series have been aggregated to daily values and afterwards disaggregated back to hourly values, which enables comparisons between observed and disaggregated time series at the same location. The main findings for hourly resolution from Müller and Haberlandt (2015) are shown in Table 7 as relative errors $r$ between disaggregated ($Dis$) and observed ($Obs$) time series over all investigated stations ($n=9$ in their case) :

$$r = \frac{1}{n} \times \sum_{i=1}^{n} \frac{(r_{Dis} - r_{Obs})}{r_{Obs}} \tag{8}$$

Slight underestimation of dry spell duration (relative error of -6 %), fraction of dry intervals (-3 %), wet spell duration (-12 %) and amount (-9 %) can be identified, while average intensity is slightly overestimated (4 %). While the autocorrelation function also shows underestimations, the extreme values are well represented (not shown here).

**Table 7. Relative error of rainfall characteristics resulting from the disaggregation model for the same investigation area from Müller and Haberlandt (2015)**

| Rainfall characteristic | Relative error r [%] |
|---|---|
| Wet spell duration [h] | -12 |
| Standard deviation | -29 |
| Skewness | -26 |
| Wet spell amount [mm] | -9 |
| Standard deviation | -18 |
| Skewness | -19 |
| Dry spell duration [h] | -6 |
| Standard deviation | -7 |
| Skewness | 9 |
| Fraction of dry intervals [%] | -3 |
| Average intensity [mm/h] | 4 |

In Fig. 5 the bivariate characteristics are shown for V1, V2 and V3 in comparison with the observations. For the V1 case (the disaggregated time series without any subsequent steps), the probability of occurrence and the correlation coefficients are underestimated, whereas the continuity ratio is overestimated.

For the V2 case, the probability of occurrence and the correlation coefficients could be improved. While values for the probability of occurrence and correlation coefficient for rainfall intensities > 4 mm are similar to observations, a slight underestimation can be identified for correlation coefficients for rainfall intensities ≤ 4 mm for some station pairs. For the





continuity ratio, V2 results are varying. This is due to the definition of the criterion, taking into account station $k$ with respect to station $l$, but not vice versa. This definition leads to different values for the same station pair, because different time steps are taken into account. Therefore, for $C_{k,l}$ an improvement can be identified during simultaneous worsening of $C_{l,k}$.

It should be noted that the resampling algorithm has not been validated in the context of distances smaller than 20 km and temporal resolution of 1 h. Although the spatial rainfall characteristics are underestimated after the disaggregation (V1), a major improvement can be identified moving all station pairs into the cloud of observations (except some of the continuity ratio).

The simultaneous rainfall of V3 leads to the best values for the continuity ratio, comparable to those from observations. However, slight overestimations can be identified for both coefficients of correlation. For the probability of occurrence, high overestimations can be identified (approximately 50 %). Although the same diurnal cycles are used for all stations, probability of occurrence is less than 1 due to the fact that rainfall does not necessarily occur at all stations on a wet day.

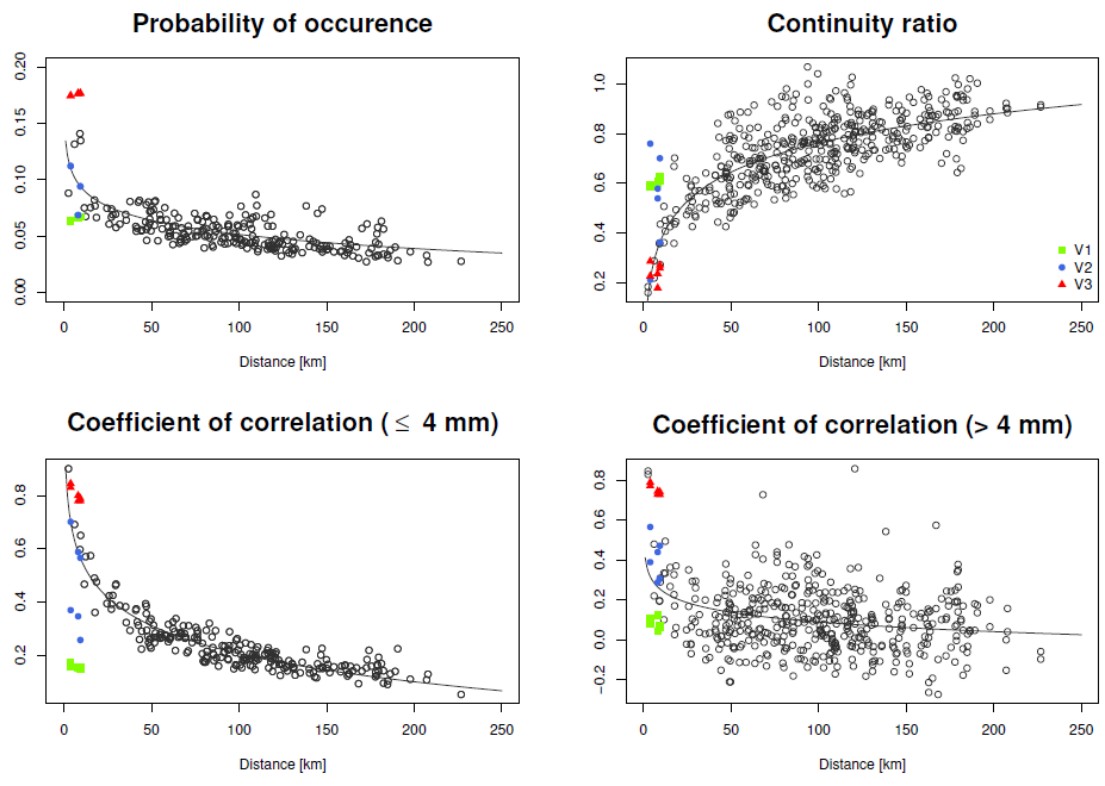

**Fig. 5. Bivariate spatial rainfall characteristics of V1, V2 and V3 in comparison to observations for the catchment Pionierbrücke (for one realization).**

Additionally, the influence of the spatial consistence on resulting areal rainfall intensities is investigated. In Fig. 6, areal rainfall intensities resulting from V1, V2 and V3 are shown for one subcatchment of Pionierbrücke. Since only one observed





high-resolution time series (Reckershausen: two) is available for each catchment, no comparison between areal rainfall intensities between observed and disaggregated time series (resulting from three stations for each catchment) can be carried out. Areal rainfall intensities resulting from disaggregated time series can only be compared among each other. V1 leads to the lowest rainfall intensities, V3 to the highest. Areal rainfall intensities of V2 lie between V1 and V2. The "random"

5    rainfall occurrence in V1 leads to smaller rainfall intensity values as was indicated by the probability of occurrence (see Fig. 5). Accordingly, the parallelization of V3 leads to the highest areal rainfall intensities. Therefore, the results for the spatial bivariate characteristics and the areal rainfall intensities are consistent. The findings are similar for the other subcatchments in Tetendorf and Reckershausen.

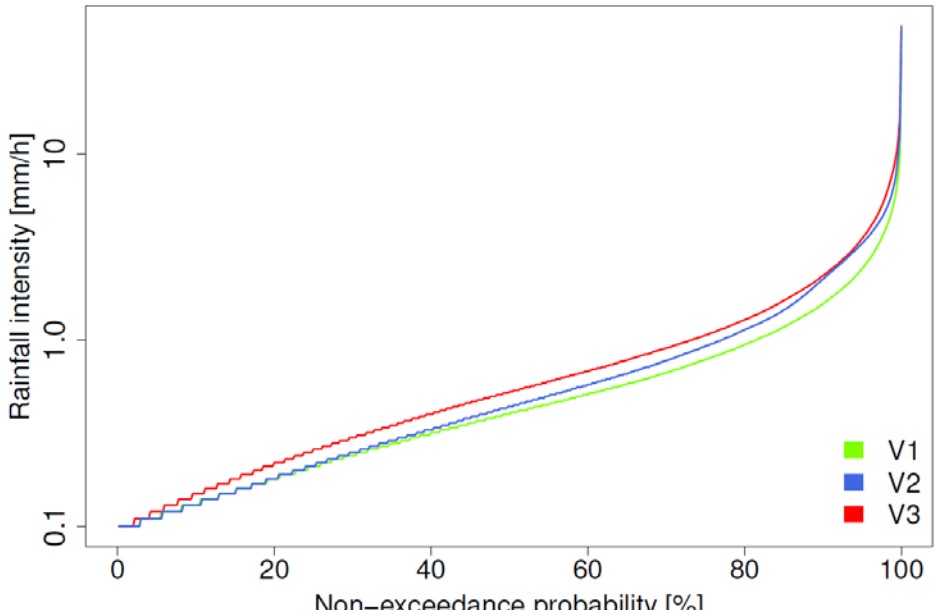

**Fig. 6. Non-exceedance curve of areal rainfall intensities for V1, V2 and V3 for one subcatchment of Pionierbrücke (for one realization)**

Additionally, the extreme values of the areal rainfall intensities have been analyzed, since those can have a significant influence on the resulting runff. In Fig. 7 the annual rainfall extremes for another subcatchment in Pionierbrücke are

15    illustrated using the Weibull-plotting position (similar for all subcatchments). As identified for all areal rainfall intensities, V1 also leads for the extreme values to the lowest values for each return period. V2 and V3 result in similar values for return periods < 18 years. The clear difference of higher values for V3 over the whole spectrum of non-exceedance probability from Fig. 6 cannot be identified for the extreme values. Although for V3, where the diurnal cycle of the station with the highest daily rainfall amount is transferred to the time series of all other stations, V3 does not lead to the highest extreme





values. The reason for this is that the highest daily rainfall amount does not necessarily lead to the highest rainfall intensity on the final disaggregation level with an hourly time step. As an example, a rainfall station A with a daily total rainfall amount of 50 mm has a maximum intensity during this day of 8 mm/h, whereas station B with a daily total rainfall of 40 mm has a higher maximum intensity of 15 mm/h. As such, V3 can also lead to a smoothing of the rainfall intensities, at least for

5    peak intensities. However, for higher return periods (> 18 years), V3 leads to higher values than V2.

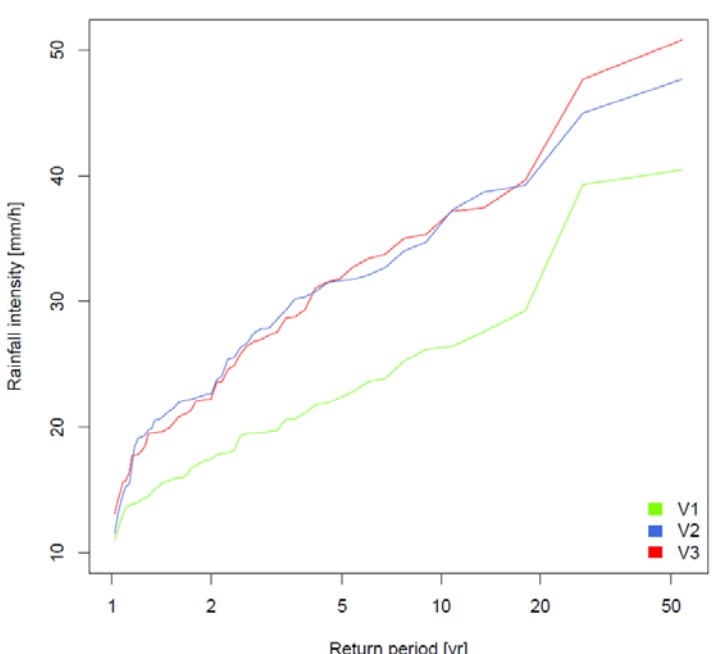

**Fig. 7. Annual rainfall extremes of the areal rainfall intensities for subcatchment 2 in Pionierbrücke (for one realization)**

It can be summarized that V1, V2 and V3 lead to different results regarding spatial characteristics and areal rainfall intensities

### 4.2 Rainfall-runoff-model results

In this chapter, all rainfall-runoff simulation results are presented. The chapter is organized as follows: in a) the rainfall runoff-model results using HBV are shown for all catchments for V1, V2 and V3 with three rain gauges as input for each. In b) HBV-model results for different station densities for catchment Reckershausen are presented. HBV-model results without

15    parameter calibration are shown for all catchments in c), while WaSiM-model results are presented in d) for catchment Pionierbrücke.



### a) HBV- simulation results with calibration using three rain gauges as input

The parameterization was carried out by a split-sampling with a calibration and validation period for each catchment. The results for Reckershausen, Pionierbrücke and Tetendorf are shown in Fig. 8, 10 and 11 for the calibration period. For Reckershausen, only results using three rain gauges as input are shown here. For Extr-Su and Extr-Wi, flood quantiles are shown for a return period of 100 years. However, the extrapolation is limited by the length of the simulated runoff time series. As per Maniak (2005), a maximum return period of three times the runoff time series length should be used to avoid too high statistical uncertainties caused by extrapolation. This results in 75 years for Pionierbrücke, 21 years for Tetendorf and 45 years for Reckershausen. The discussion of the results is limited to those and more frequent return periods. For a quantitative analysis, *NSC*-values for all criteria and for each catchment are given in Table 8. As mentioned before, *NSC*-values are based on a few supporting points (see Eq. 6). Also, theoretical Gumbel-distribution functions with two parameters are compared, which can be similar although the population used of each distribution function are different. Hence, values of 0.99 or even 1.00 can be achieved. On the other hand, small deviations from the observations can lead to even negative *NSC*-values (see e.g. the discussion of the simulation results for Reckershausen).

For Reckershausen, the Extr-Su and Extr-Wi are similar to those from observations (Fig. 8). While for summer all observed flood quantiles are within the range of Extr-Su ($0.99 \leq NSC \leq 1.00$), for Extr-Wi a slight overestimation occurs for V2 and V3.

For the validation period, flood quantiles for both, Extr-Su and Extr-Wi, are overestimated. The overestimation is higher in winter (approx. 20 m³/s for $HQ_{50}$) than in summer (approx. 10 m³/s). One possible cause can be the higher yearly maximums in the calibration period. It is assumed that parameters, calibrated to achieve high floods, tend to generate larger discharges even if lower yearly maxima are observed. This is also indicated by the results for FDC and Q-mon. Although both are represented well in the calibration period ($0.88 \leq NSC_{FDC} \leq 0.90$, $0.96 \leq NSC_{Q\text{-}mon} \leq 0.99$), both criteria are overestimated in the validation period ($0.57 \leq NSC_{FDC} \leq 0.63$, $0.81 \leq NSC_{Q\text{-}mon} \leq 0.89$). In the validation period the range and hence the uncertainty for both, Extr-Su and Extr-Wi, is smaller for V2 and V3 in comparison to V1.

The simulation results of Extr-Su of the validation period for the catchment Reckershausen show the sensitivity of the *NSC* as a goodness-of-fit criterion. V1 and V3 lead to positive *NSC*-values (0.60 and 0.31), while V2 leads to a negative value of *NSC*=-0.05. However, from a visual inspection (see Fig. 9), differences between all three approaches are small and less intense as one might expect from the *NSC*-value itself. The high sensitivity of the *NSC* makes a direct interpretation of its values more difficult (Schaefli and Gupta, 2007, Criss and Winston, 2008). However, for the calibration process, a high sensitivity leads to an improvement of the simulation results.

Values for the objective function are given in Tab. 9. For Reckershausen, the objective function values are very similar for V1, V2 and V3 for both, calibration and validation period. Especially by taking into account that the value for the objective function depends on four *NSC*-values.





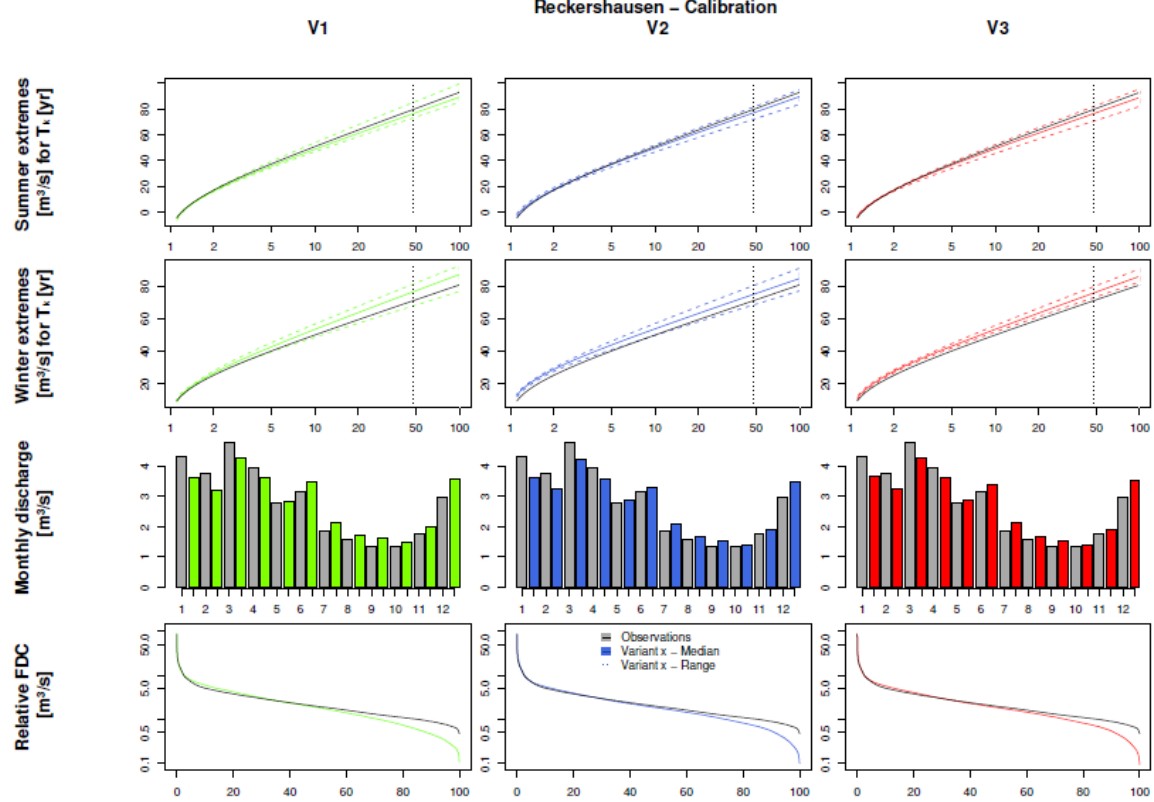

**Fig. 8. Runoff simulation results with HBV for Reckershausen, calibration period**



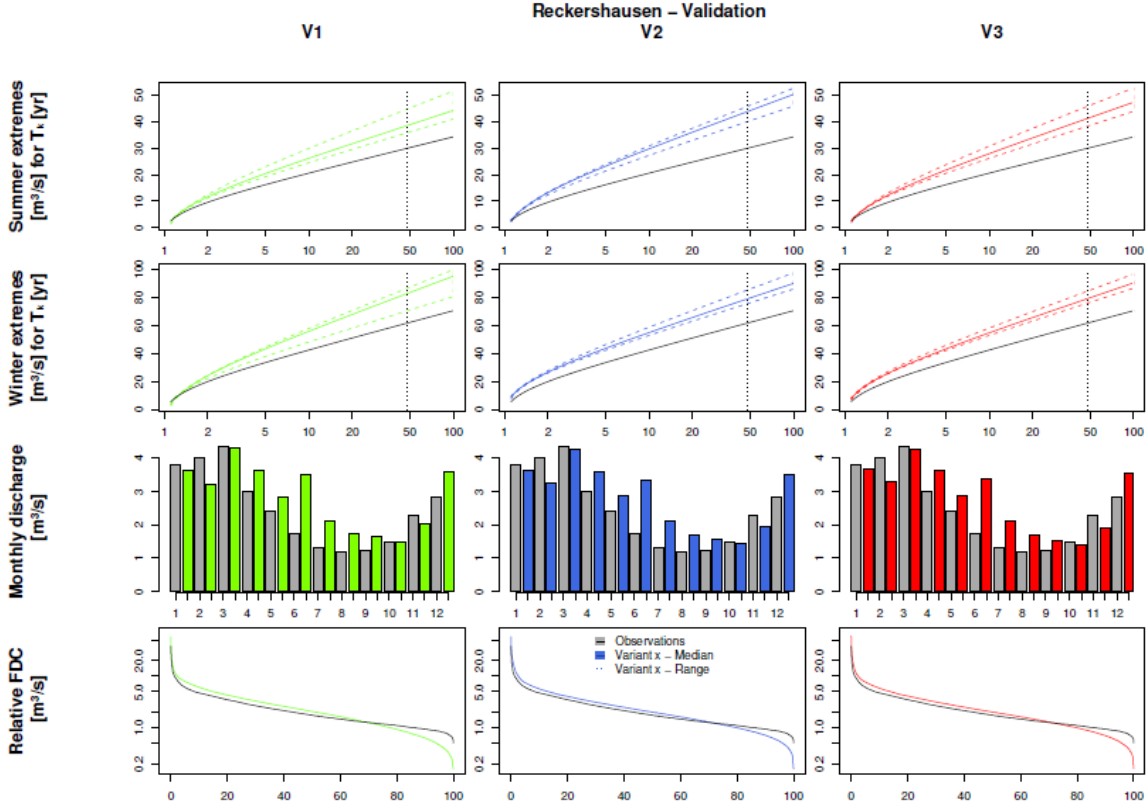

**Fig. 9. Runoff simulation results with HBV for Reckershausen, validation period**

For Pionierbrücke it should be mentioned that at points during the calibration (see the FDC in Fig. 10) and validation periods
a simulated discharge of $Q = 0$ m³/s was obtained. Zero discharge implies that all storages have been emptied. This occurs

5    only for Pionierbrücke and is due to the very steep conditions in the mountainous catchment (see Fig. 1) and hence the low
soil depth and storage capacity. In the observed time series the minimum value is $Q = 0.1$ m³/s. The underestimation is as
well caused by the selection of criteria selected for the objective function used for calibration. The main aim is to represent
the extreme flows, while the shape of the intra-annual cycle of monthly average discharges and of the FDC are only
implemented to achieve an overall realistic mean discharge behavior. For the FDC, four quantiles greater than 0.5 and only

10   two quantiles smaller than 0.5 are used. Smaller quantiles are not of interest in these simulations, since discharge values in
that range belong to dry periods with low flows, for which daily values of rainfall are sufficient for simulations and hence no
rainfall disaggregation would be necessary. For the FDC, V3 leads to a slightly better fit to observations for non-exceedance
probabilities smaller than 35 %, but to a worse fit between 35 % and 60 % non-exceedance probability. However, FDC is
underestimated, independent from the applied rainfall data set, for non-exceedance probabilities higher than 60 %. The





underestimation identified by the FDC can also be identified for Q-mon in winter and in the underestimation of the Extr-Su and Extr-Wi. The results for the validation period are very similar and not shown here.

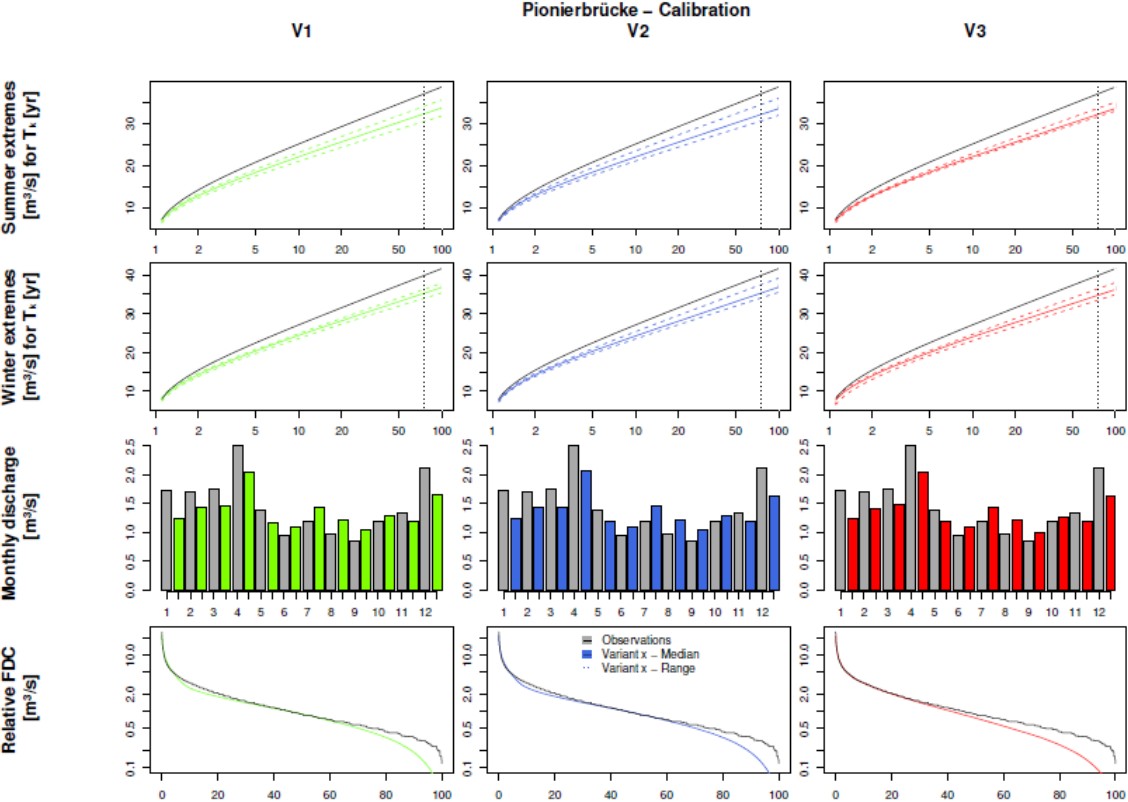

**Fig. 10. Runoff simulation results with HBV for Pionierbrücke, calibration period**

On the contrary, for Tetendorf FDC and Q-mon (except September and October) are overestimated by all rainfall data sets (Fig. 11). However, for Q-mon the shape of the intra-annual cycle is well-represented. For the extreme values it should be mentioned again, that the analyzes are only valid for return periods more frequent than 21 years. For Extr-Su, underestimations occur for return periods more frequent than 5 years for all variants in the calibration period (less than 2 years in the validation period). For Extr-Wi, the median of V1 represents the observed values well, while for V2 and V3 the median leads to overestimations for return periods frequent than 5 years. However, observations are still in the range of the simulation results, whereby the range is wider for V1 and V3 in comparison to V2. In total, the resampling in V2 leads to a reduction of the overestimation of the observed summer extreme values, but to a stronger overestimation for winter extremes in comparison to V1 and V3.



Since for Tetendorf seasonal differences regarding V2 were identified, the spatial rainfall characteristics of the objective function applied for the resampling process have been re-analyzed, differing between the summer and winter half years. The results regarding both periods as well as the estimation over the complete year are shown in Fig. 12 for all bivariate spatial rainfall characteristics. For the continuity ratio, probability of occurrence and both volume classes of correlation coefficients,

differences can be identified, based on the different geneses of rainfall in summer and winter. The probability of rainfall occurrence is lower in summer due to a higher amount of convective rainfall events. However, the distance-dependent curve progression is very similar between the seasonal and annual estimated spatial characteristics. Since spatial characteristics are just moved closer to the regression line by V2 (without a perfect fit, see Fig. 5), an improvement of the spatial rainfall characteristics by introducing slightly different season-dependent regression lines cannot be expected and is hence not

applied.

As main reasons for the seasonal differences, the short validation and calibration period are considered. Short periods mean a small amount of days with rain and hence a small amount of relative diurnal cycles to swap during the resampling, limiting the ability of the algorithm to improve the spatial characteristics. The usage of time series of V2 as input for HBV and the additional short time for the calibration process lead to the seasonal differences.

For longer calibration and validation periods (Reckershausen and Pionierbrücke) the results for V1, V2 and V3 are very similar regarding the runoff statistics. An influence of the chosen method for the implementation of spatial consistence cannot be recognized.

**Table 8.** *NSC*-values for all catchments and all criteria for calibration (Cal) and validation (Val) period

| Catchment | Criteria | V1 | | V2 | | V3 | |
|---|---|---|---|---|---|---|---|
| | | Cal | Val | Cal | Val | Cal | Val |
| Reckershausen | *Extr-Su* | 0.99 | 0.60 | 1.00 | -0.05 | 0.99 | 0.31 |
| | *Extr-Wi* | 0.97 | 0.43 | 0.97 | 0.58 | 0.97 | 0.58 |
| | *FDC* | 0.88 | 0.57 | 0.90 | 0.63 | 0.90 | 0.61 |
| | *Q-mon* | 0.96 | 0.81 | 0.99 | 0.89 | 0.98 | 0.85 |
| Pionierbrücke | *Extr-Su* | 0.89 | 0.95 | 0.88 | 0.91 | 0.89 | 0.94 |
| | *Extr-Wi* | 0.91 | 0.88 | 0.91 | 0.86 | 0.89 | 0.83 |
| | *FDC* | 0.61 | 0.17 | 0.61 | 0.16 | 0.61 | 0.17 |
| | *Q-mon* | 0.99 | 1.00 | 0.99 | 1.00 | 0.99 | 0.99 |
| Tetendorf | *Extr-Su* | 0.32 | -0.79 | 0.68 | 0.78 | 0.21 | -0.61 |
| | *Extr-Wi* | 0.87 | 0.70 | 0.64 | -4.36 | 0.47 | 0.88 |
| | *FDC* | 0.79 | 0.82 | 0.84 | 0.65 | 0.71 | 0.78 |
| | *Q-mon* | 0.86 | 0.93 | 0.78 | 0.92 | 0.83 | 0.92 |




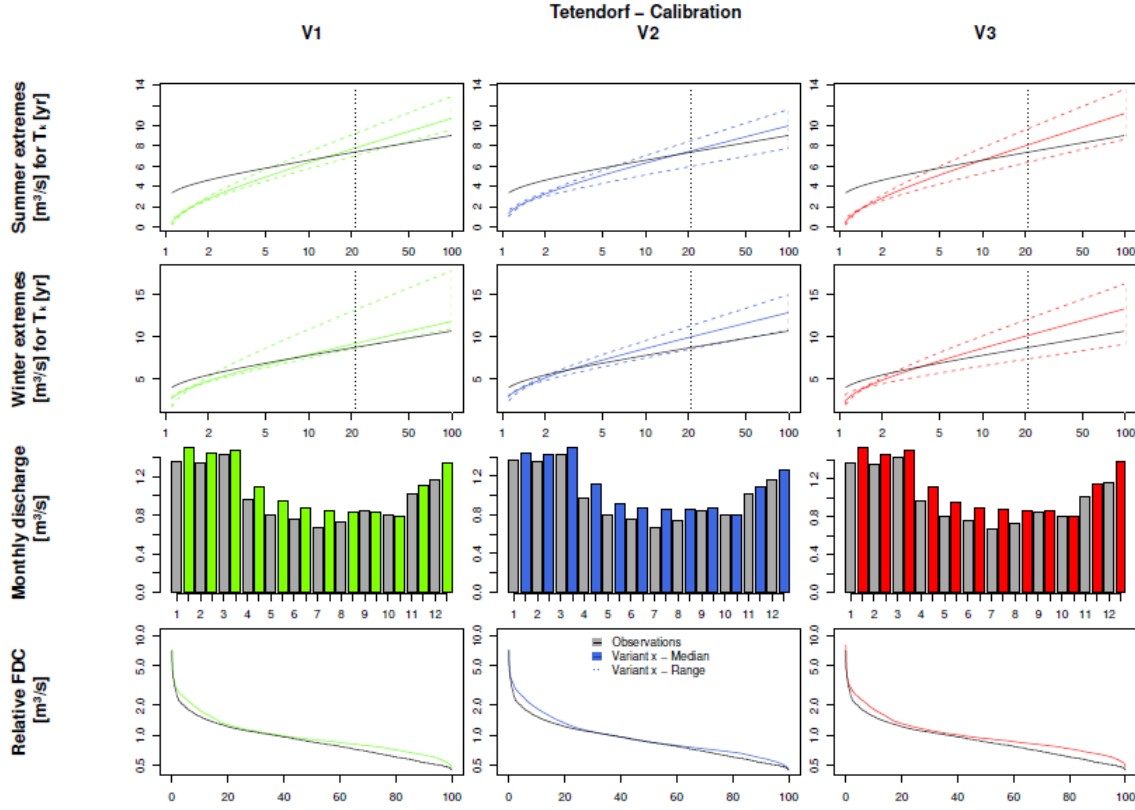

**Fig. 11. Runoff simulation results with HBV for Tetendorf, calibration period**

**Table 9. $O_{stat}$-values for all catchments and all criteria for calibration (Cal) and validation (Val) period**

| Catchment | V1 | | V2 | | V3 | |
| --- | --- | --- | --- | --- | --- | --- |
| | **Cal** | **Val** | **Cal** | **Val** | **Cal** | **Val** |
| Reckershausen | 0.04 | 0.39 | 0.03 | 0.48 | 0.03 | 0.40 |
| Pionierbrücke | 0.13 | 0.21 | 0.13 | 0.23 | 0.14 | 0.23 |
| Tetendorf | 0.29 | 0.58 | 0.27 | 1.49 | 0.44 | 0.50 |





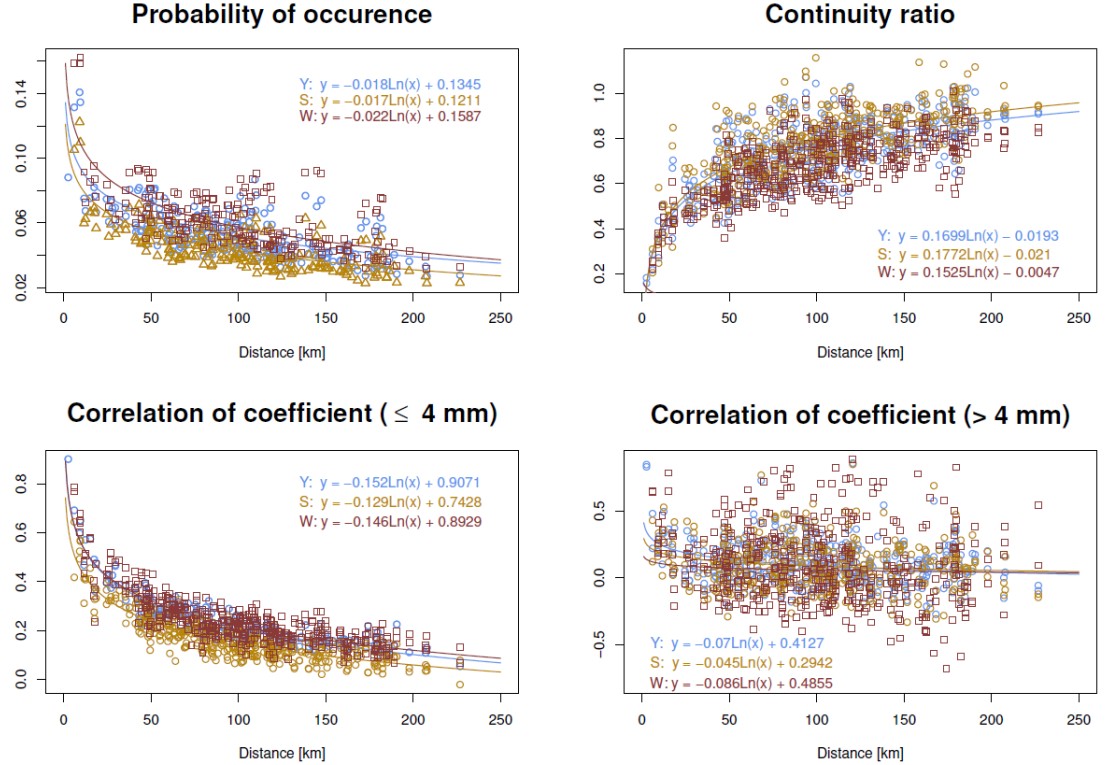

**Fig. 12. Bivariate spatial characteristics estimated for summer (S) and winter (W) seasonal as well as over the whole year (Y)**

**b) HBV- simulation results calibration using different numbers of rain gauges as input**

A possible reason for the non-visible influence of the chosen method for the implementation of spatial consistence in the simulated runoff statistics is the low rain gauge network density. With a low network density, it is not possible to reflect the spatial rainfall variability and hence the influence of V1, V2 and V3 cannot be identified. The influence of the spatial rainfall variability on the runoff can only be determined by rainfall-runoff simulations. Investigations to the influence of spatial rainfall patterns can be found in Krajewski et al. (1991), Ogden and Julien (1993), Obled et al. (1994) and Nicotina et al. (2008).

Therefore, for Reckershausen, different numbers of rain gauges are applied for the calculation of the areal rainfall used as input for HBV. Areal rainfall is estimated by 3 rain gauges (representing a network density of 0.9 gauges per 100 km²) as carried out in a), 5 (1.6 gauges/100 km²) and 8 rain gauges (2.5 gauges/100 km²). The results are shown for V2 in Fig. 13 for the calibration and in Fig. 14. for the validation period. The results for V1 and V3 are very similar and not shown here. However, for a quantitative analysis the *NSC*- and $O_{stat}$-values are shown in Table 10 and Table 11.



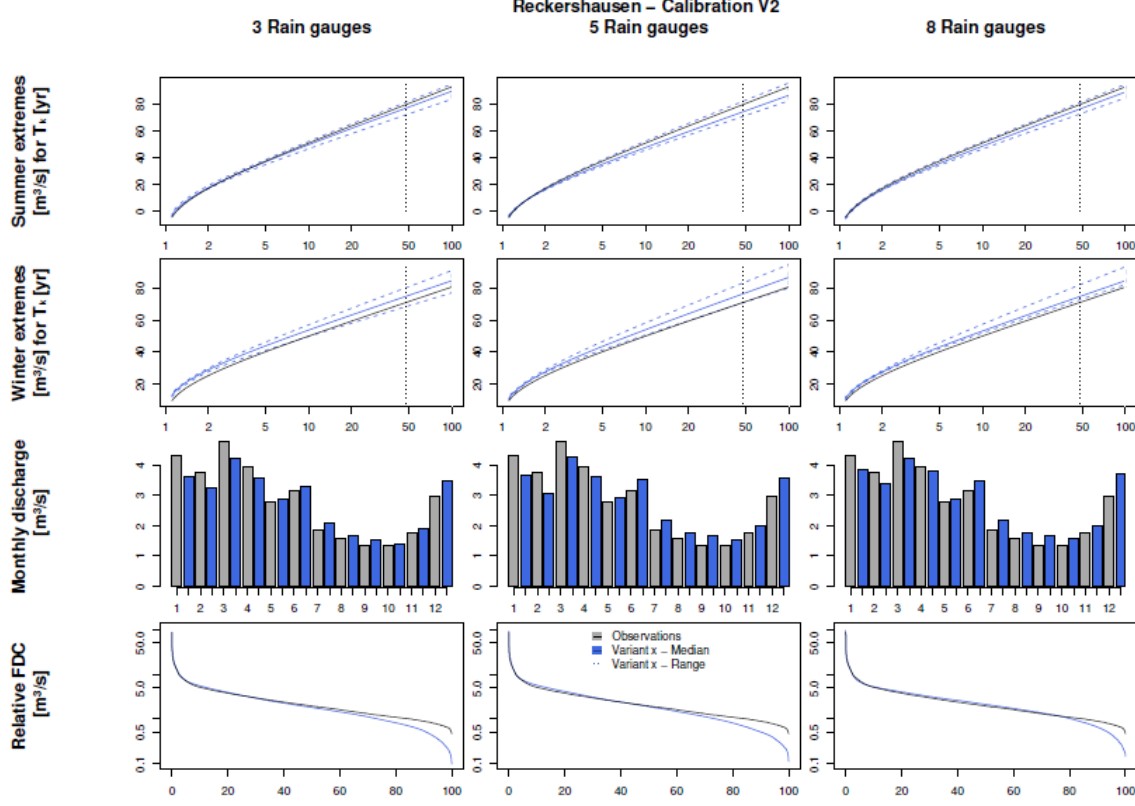

**Fig. 13. Runoff simulation results for V2 with 3, 5 and 8 rain gauges with HBV for Reckershausen, calibration period**



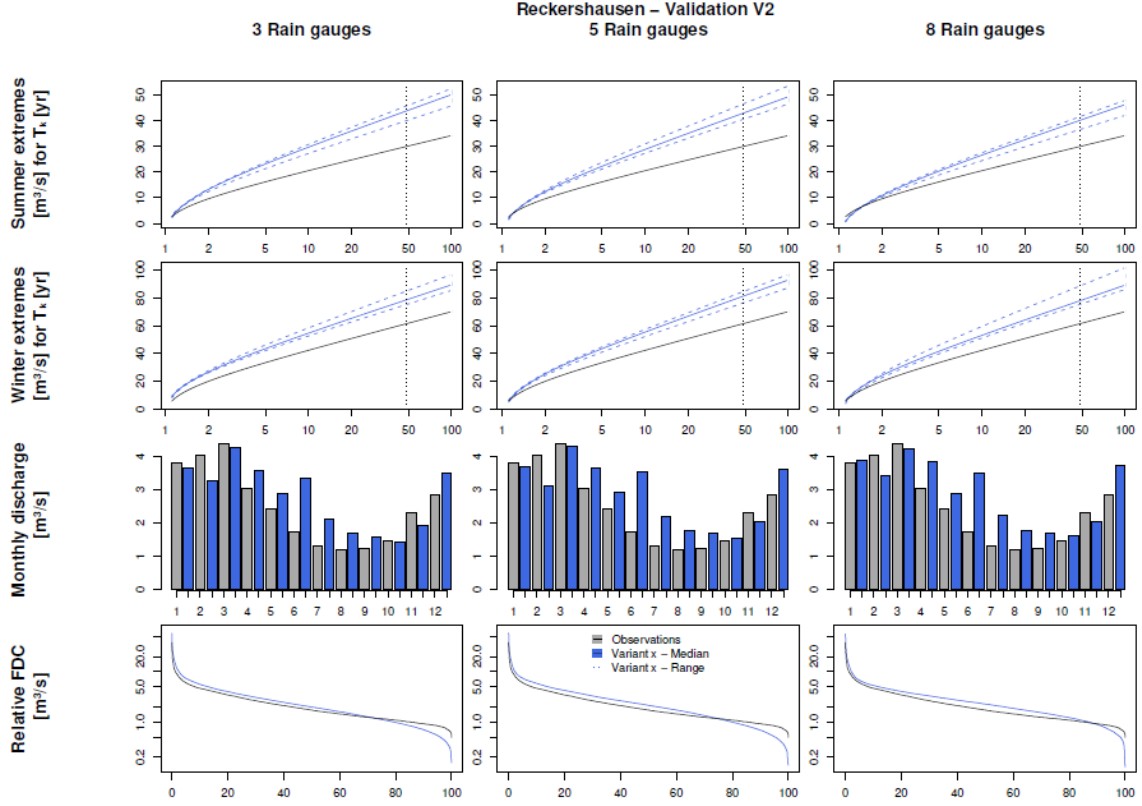

**Fig. 14. Runoff simulation results for V2 with 3, 5 and 8 rain gauges with HBV for Reckershausen, validation period**

Again, independent of the number of rain gauges used for the estimation of the areal rainfall, the results from the calibration period (Fig.13) represent the observations better than those from the validation period (Fig. 14). In the validation period, Extr-Su and Extr-Wi are overestimated as well as the majority of Q-mon and the FDC. Minor differences can be identified between the different rain gauge network densities, but no general conclusion is possible, e.g. the overestimation of Extr-Wi in the calibration period is increasing with an increasing network density. However, in the validation period the overestimation is decreasing with an increasing number of rain gauges from 3 to 8. Also for Q-mon or the FDC, no systematic improvement can be identified. This is an unexpected finding, because with the additional information from the daily total rainfall amounts, an improvement of at least the continuum characteristics was expected. Also for the *NSC-* and $O_{stat}$-values no systematical improvement can be identified: $O_{stat}$(V2, 3 rain gauges)= 0.03, $O_{stat}$(V2, 5 rain gauges)= 0.04, $O_{stat}$(V2, 8 rain gauges)= 0.03 (see Table 10 and Table 11).

It can be summarized, that the number of rain gauges has only a minor, but no systematic influence on runoff statistics for the catchments used in this investigation. Investigated number of rain gauges were 3, 5 and 8, respectively 0.9, 1.6 and 2.5 rain gauges/100 km². This contradicts conclusions from other studies. Seliga et al. (1992) recommend for spatial rainfall





applications information every 5 km² (20 rain gauges/ 100 km²). So an improvement by an increasing station density up to this threshold should have been expected. For a French catchment with an area size of 71 km², Obled et al. (1994) investigated the influence of using 5 or 21 rain gauges, representing rain gauge network densities of 7 and 22 rain gauges/100 km². With 21 rain gauges Obled et al. improved their results significantly. Nevertheless, they conclude that the improvement is based on the better estimation of the total rainfall amount, not on its spatial distribution. Xu et al. (2013) investigated the influence of station density on a Chinese catchment with an area size of 94 660 km² and daily rainfall time series, hence a direct comparison of network densities is not possible. Nevertheless, they point out that the distribution of rain gauges inside the catchment is of importance. A distribution covering regions with different rainfall behaviors in a catchment can lead to better simulation results with only a few rain gauges in comparison to a less efficiently distributed network with more rain gauges. In the actual study, the rain gauges for each network density scenario have been selected in a way to cover the catchment area and its rainfall representatively (see Fig. 2). This could be one reason why an increase in rain gauge network density shows no systematic improvement in this study.

**Table 10.** *NSC*-values for all catchments and all criteria for calibration (Cal) and validation (Val) period

| Number of rain gauges | Criteria | V1 | | V2 | | V3 | |
|---|---|---|---|---|---|---|---|
| | | Cal | Val | Cal | Val | Cal | Val |
| 3 | *Extr-Su* | 0.99 | 0.6 | 1 | -0.05 | 0.99 | 0.31 |
| | *Extr-Wi* | 0.97 | 0.43 | 0.97 | 0.58 | 0.97 | 0.58 |
| | *FDC* | 0.88 | 0.57 | 0.9 | 0.63 | 0.9 | 0.61 |
| | *Q-mon* | 0.96 | 0.81 | 0.99 | 0.89 | 0.98 | 0.85 |
| 5 | *Extr-Su* | 0.98 | -0.24 | 0.98 | 0.09 | 0.99 | -0.23 |
| | *Extr-Wi* | 0.97 | 0.68 | 0.96 | 0.48 | 0.98 | 0.65 |
| | *FDC* | 0.86 | 0.53 | 0.87 | 0.53 | 0.86 | 0.55 |
| | *Q-mon* | 0.99 | 0.91 | 0.98 | 0.86 | 0.99 | 0.91 |
| 8 | *Extr-Su* | 0.99 | 0.75 | 0.99 | 0.46 | 1 | 0.54 |
| | *Extr-Wi* | 0.96 | 0.62 | 0.98 | 0.64 | 0.97 | 0.59 |
| | *FDC* | 0.91 | 0.57 | 0.89 | 0.54 | 0.89 | 0.6 |
| | *Q-mon* | 0.99 | 0.88 | 0.99 | 0.94 | 0.98 | 0.88 |





**Table 11.** $O_{stat}$-values for all catchments and all criteria for calibration (Cal) and validation (Val) period

| Number of | V1 | | V2 | | V3 | |
|---|---|---|---|---|---|---|
| rain gauges | Cal | Val | Cal | Val | Cal | Val |
| 3 | 0.04 | 0.39 | 0.03 | 0.48 | 0.03 | 0.40 |
| 5 | 0.05 | 0.51 | 0.04 | 0.49 | 0.04 | 0.51 |
| 8 | 0.04 | 0.28 | 0.03 | 0.34 | 0.04 | 0.33 |

**c1) HBV-simulation results without calibration using three rain gauges as input**

Another possible reason for the small differences between V1, V2 and V3 is the calibration of the rainfall-runoff model

parameters for each of the rainfall data sets. Parameters are allowed to vary between V1, V2 and V3, and hence damp the effects of the different degrees of spatial consistence. To exclude the calibration as a possible reason for the damping behavior, a calibration with a neutral rainfall data set offering the same spatial rainfall coverage without giving preference to one of the investigated versions would be recommended. This would enable a direct comparison between V1, V2 and V3 without re-calibration of the models. Since high-resolution time series do not exist with the required spatial network density,

radar data could be a possible solution. However, radar time series exist only for lengths which are too short for model simulations and subsequent derived flood frequency analyses.

To avoid re-calibrations, a pragmatic solution is chosen. The arithmetic means of the lower and upper limit for each parameter in Table 5 have been applied as a neutral parameter set. For the validation period simulation results based on this neutral parameter set have been analyzed. Although a splitting in calibration and validation period is not necessary if no

calibration is carried out, comparisons are possible between the simulation results with and without calibrated parameters. The results are shown in Fig. 15 for Reckershausen, results are similar for Pionerbrücke and Tetendorf. For a quantitative evaluation *NSC*-values for all catchments are listed in Table 11 and $O_{stat}$-values in Table 12.

For Pionierbrücke and Tetendorf simulation results are worse without calibration (e.g. for Pionierbrücke, V1: $O_{stat,not calibrated}$=1.14 and $O_{stat,calibrated}$=0.21). For Reckershausen a slight improvement can be identified without calibration. The

calibrated parameters led in the validation period to an overestimation of extreme values for both seasons as well as an overestimation of FDC and Q-mon (e.g. for V3: $O_{stat,not calibrated}$=0.28 and $O_{stat,calibrated}$=0.40). For all catchments, Extr-Su are underestimated by every version of spatial consistence. The Extr-Wi are also underestimated for Reckershausen and Pionierbrücke, but overestimated for Tetendorf. For all catchments, an intra-annual cycle of Q-mon can be identified. For Reckershausen, Q-mon is similar to observations, while for Pionierbrücke underestimations and for Tetendorf

overestimations can be identified in winter. The FDC is not represented well for any of the catchments.

Although a neutral set of parameters has been applied, the differences in the simulation results between V1, V2 and V3 are still small. For Pionierbrücke the values of the objective function show the same range without and with calibration



$(1.10 \text{ (V2)} \leq O_{stat, not\ calibrated} = \leq 1.14 \text{ (V1)}$ respectively $0.21 \text{ (V1)} \leq O_{stat, calibrated} \leq 0.23 \text{(V2, V3))}$. The similarity of the simulation results exists even if the model parameters are not calibrated and a neutral parameter set is used.

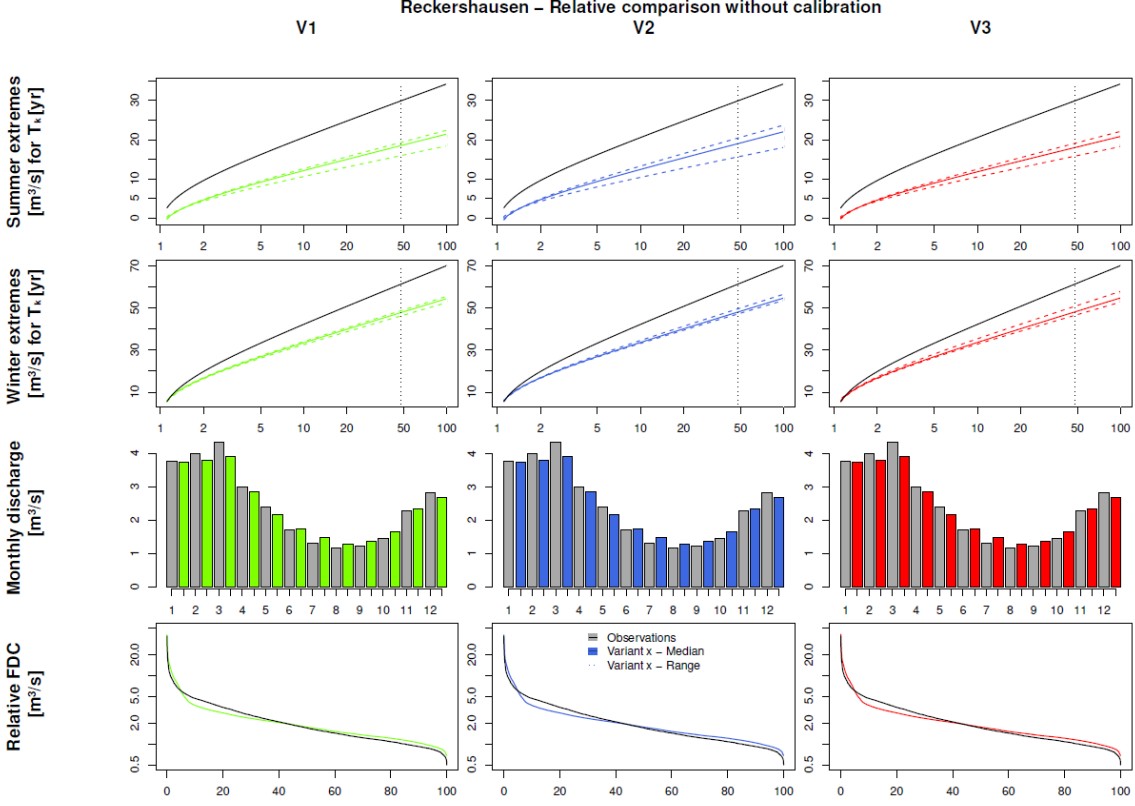

**Fig. 15. Runoff simulation results with HBV without calibration for Reckershausen, validation period**



**Table 12. NSC-values for all catchments and all criteria without calibration for validation period**

| Catchment | Criteria | V1 | V2 | V3 |
|---|---|---|---|---|
| Reckershausen | *Extr-Su* | 0.20 | 0.26 | 0.14 |
| | *Extr-Wi* | 0.76 | 0.77 | 0.77 |
| | *FDC* | 0.97 | 0.97 | 0.97 |
| | *Q-mon* | 0.99 | 0.99 | 0.99 |
| Pionierbrücke | *Extr-Su* | -1.68 | -1.58 | -1.59 |
| | *Extr-Wi* | 0.01 | 0.10 | 0.06 |
| | *FDC* | -0.07 | -0.07 | -0.07 |
| | *Q-mon* | 0.96 | 0.96 | 0.96 |
| Tetendorf | *Extr-Su* | 0.52 | 0.54 | 0.55 |
| | *Extr-Wi* | -7.78 | -7.78 | -8.41 |
| | *FDC* | 0.19 | 0.19 | 0.19 |
| | *Q-mon* | -0.06 | -0.05 | -0.11 |

**Table 13. $O_{stat}$-values for all catchments and all criteria without parameter calibration for validation period**

| Catchment | V1 | V2 | V3 |
|---|---|---|---|
| Reckershausen | 0.27 | 0.25 | 0.28 |
| Pionierbrücke | 1.14 | 1.10 | 1.11 |
| Tetendorf | 2.79 | 2.79 | 2.96 |

**c2)       WaSiM- simulation results without calibration using three rain gauges as input**

For the comparison of V1, V2 and V3, WaSiM (Schulla, 1997, 2015) as a physically based and distributed hydrological
model is used as an additional rainfall-runoff model. The application of more than one model increases the reliability of the
10  simulation results and excludes the possibility of being model-dependent. As mentioned before, a pre-calibration of the
rainfall-runoff model was carried out focusing only on the water balance itself and not on the objectives used in Eq. (7).
Only the parameters mentioned in the model description (see 3.2.b) are calibrated. As far as possible, the same parameter
values as in HBV in the uncalibrated case (c1) have been applied. The investigation with WaSiM is carried out only for
catchment Pionierbrücke, since here the highest differences in simulation results are expected due to the short reaction time
15  of the catchment.





The results are shown in Fig. 16 for the calibration period, Fig. 17 for the validation period and a quantitative analysis is given in Table 14. For the calibration and the validation period Extr-Su and Extr-Wi are simulated slightly higher with V2 and V3 in comparison to V1. This is consistent with the areal rainfall extremes presented for Pionierbrücke in Fig. 7. In addition, the range for both criteria is higher for V2 and V3 in comparison to V1, whereby V2 leads to even wider ranges

5 than V3 in some cases (e.g. Extr-Win the validation period). In this context it should be repeated, that a relative comparison is carried out and under- or overestimations are not points of interest. The *NSE*-values for both Extr-Su and Extr-Wi are very similar for V2 and V3 (e.g. $NSC_{Extr-Wi,Cal,V2}$=0.98and $NSC_{Extr-Wi,Cal,V3}$=0.99), but show differences to V1 ($NSC_{Extr-Wi,Cal,V1}$=0.90). Hence, in WaSiM a slight effect of the spatial consistence of rainfall is visible from the simulation results. Possible reasons for the differences are the spatial resolution (150 m x 150 m for each raster cell) and the IDW algorithm

10 with an altitudinal rainfall adjustment, which was carried out by a linear regression model. However, for FDC and $Q_{mon}$, values for V1, V2 and V3 are again very similar. While for the calibration period the $O_{stat}$-values are similar for all rainfall data sets, in the validation period the $O_{stat}$-values for V2 and V3 ($O_{stat, Val, V2}$=0.45 and $O_{stat, Val, V3}$=0.46) are much closer to each other than to V1 ($O_{stat, Val, V1}$=0.30).

**Table 14.** *NSC*- and $O_{stat}$-values for Pionierbrücke without parameter calibration using WaSiM

| Criteria | V1 | | V2 | | V3 | |
|----------|------|-------|------|-------|------|-------|
| | Cal | Val | Cal | Val | Cal | Val |
| *Extr-Su* | 0.95 | 0.96 | 0.97 | 0.95 | 0.96 | 0.95 |
| *Extr-Wi* | 0.90 | 0.77 | 0.98 | 0.21 | 0.99 | 0.26 |
| *FDC* | 0.86 | -0.15 | 0.87 | -0.20 | 0.88 | -0.27 |
| *Q-mon* | 0.99 | 0.99 | 0.99 | 0.99 | 1.00 | 0.99 |
| $O_{stat}$ | 0.07 | 0.30 | 0.04 | 0.45 | 0.04 | 0.46 |



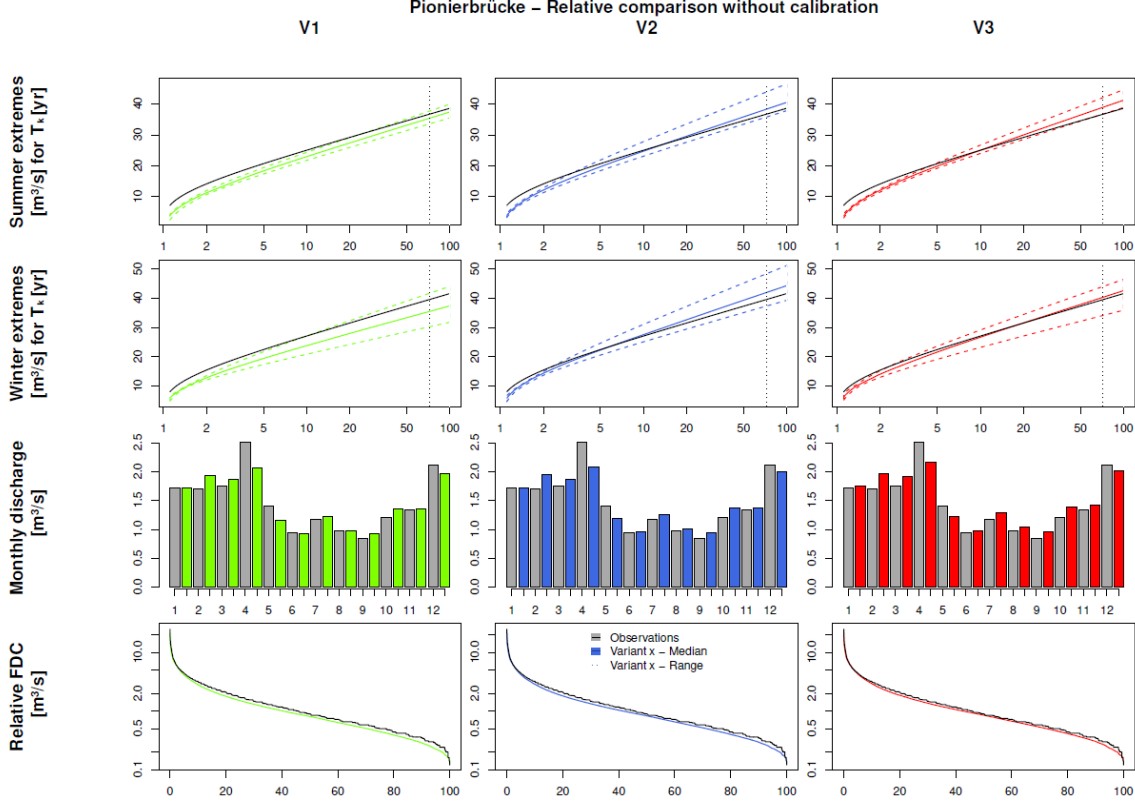

**Fig. 16. Runoff simulation results with WaSiM without calibration for Pionierbrücke, calibration period**





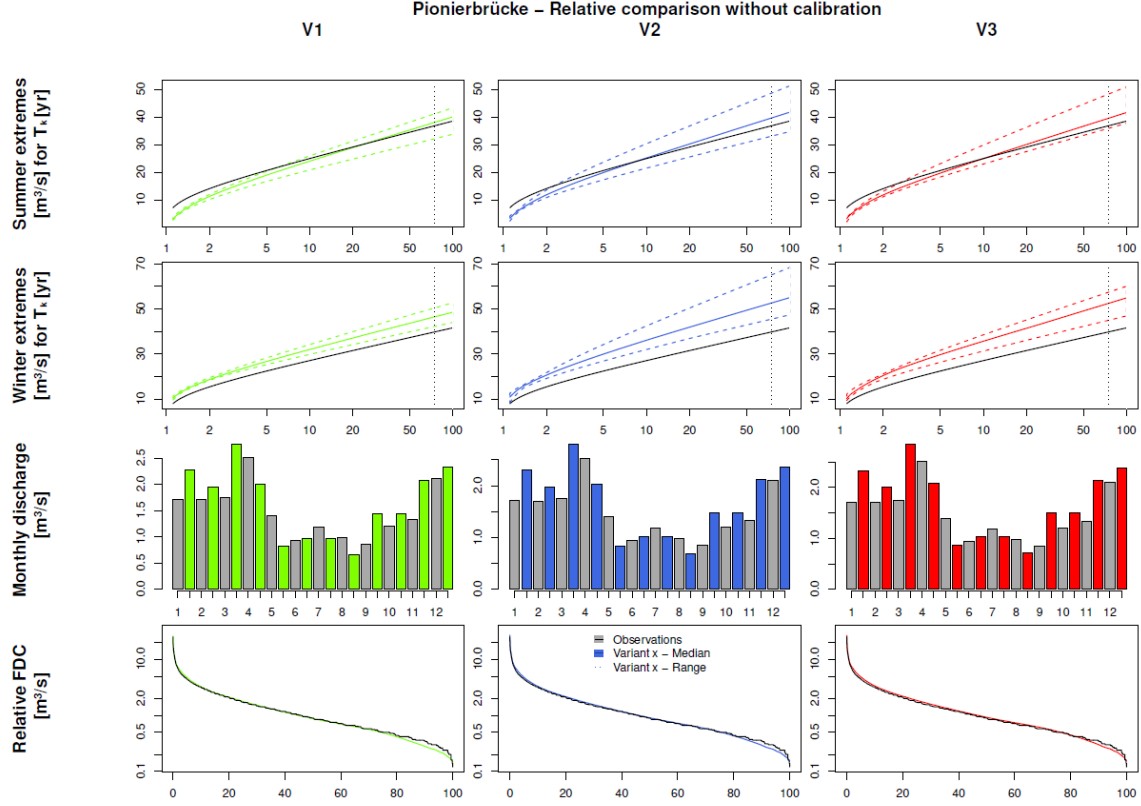

**Fig. 17. Runoff simulation results with WaSiM without calibration for Pionierbrücke, validation period**





## 5. Discussion of rainfall-runoff simulation results

The rainfall-runoff simulation results with HBV after calibration of the parameters show that with all three rainfall data sets, V1, V2 and V3, the Extr-Su and Extr-Wi, the FDC, and Q-mon can be represented with a comparable quality. The differences between the three methods are very small for the majority of all cases. Possible reasons for these small

differences, which are discussed below, are:

- small differences between the three rainfall data sets

- dampening of those differences by the calibration of the rainfall-runoff model parameters

- dampening behavior of the catchments

- choice of the rainfall-runoff model and its ability to represent differences of the three rainfall data sets

Small differences between V1, V2 and V3 would lead to small differences in rainfall-runoff simulation results. However, the differences between the three methods are apparent. For the bivariate spatial characteristics (Fig. 5), the areal rainfall intensities (Fig. 6) and the areal rainfall extremes (Fig. 7), differences can be identified among all three methods, which should as well be reflected by the runoff statistics results.

Another cause can be the separate calibration of the rainfall-runoff model parameters for each method. The applied

calibration strategy has the capability to harmonize the different rainfall data sets with the runoff statistics used for calibration. For the discussion of this harmonization effect, the simulation results for Reckershausen during the calibration (Fig. 13) and validation periods (Fig. 14) are used. During the calibration period, higher values for Extr-Su and Extr-Wi can be found in the observed runoff data. Hence, the parameters calibrated in this period tend to lead to higher runoff values. This is proven by the simulation results of the validation period with an overestimation of all runoff statistics. Only by the

usage of an uncalibrated parameter set the calibration can be excluded from the list of possible causes.

The dampening behavior of the investigated catchments depends on the size and the concentration time of a catchment (Andres-Domenech et al., 2015). Also, catchments act as a filter, so that rainfall as an input signal is dampened during its transformation to runoff by several processes (e.g. interception, losses due to storage filling, transport processes). Mandapaka et al. (2009) have analyzed for (sub-) catchments of different sizes the runoff response from different rainfall

scenarios with a total amount of 10 mm. For catchments with an area less than 10 km², a strong dependence of the duration, the intensity and the spatial distribution of the rainfall is identified. With increasing area size, the influence of these factors is reduced and for catchments with 1000 km², it is almost completely dampened. Since the catchment areas in the actual study range between 44 km² and 321 km², i.e. considerably larger than 10 km², this could be a possible reason why the differences in the runoff results are so small. On the other hand, the results of Seliga et al. (1992) and Obled et al. (1994) show that an

increasing station network density lead to an improvement of rainfall information and hence should also lead to an improvement of the runoff simulation results. Ogden and Julien (1993) investigate the time of concentration of a catchment as an influencing factor for the rainfall-runoff processes. If the duration of a rainfall event causing flooding is shorter than the time of concentration, the spatial distribution of the rainfall is influencing the discharge at the catchment outlet. If rainfall





events last longer than the concentration time, the influence decreases. However, Nicotina et al. (2008) identify an influence of spatial rainfall patterns only for catchments with areas > 1000 km², based on the travel time in the catchment. In the investigated catchments, the concentration time ranges from 1.8 h to 7.4 h, so the temporal and spatial variation should have an influence on the simulated discharges.

Another reason could be the choice of the rainfall-runoff model. Obled et al. (1994) raise the question if it is possible with semi-distributed models to transfer the information of the spatial rainfall patterns into the simulated discharge time series. Obversely, if spatial rainfall patterns are necessary for rainfall-runoff simulations for a catchment with an area size of 71 km², as is used in their study, the spatial resolution of semi-distributed models may not be sufficient. Krajeski et al. (1991) also conclude that for the analysis of spatial problems, fully-distributed models may be more suitable and recommend
those for further studies. Bárdossy and Das (2008) point out that with an increasing spatial resolution of the applied rainfall-runoff model, the sensitivity of for example the rain gauge density and hence the spatial rainfall patterns may increase as well. The rainfall-runoff simulations were carried out with two models, the semi-distributed HBV model and the fully-distributed WaSiM-model. The spatial resolution is in WaSiM with 150 m x 150 m for each raster cell much higher than in HBV with approx. 20 km² per subcatchment. This higher spatial rainfall diversity and hence a numerical diffusion of the
rainfall due to a too-coarse spatial resolution is thus avoided. Through the rainfall correction for altitude, an additional increase of the spatial diversity is achieved. While for the simulated discharge time series with HBV almost no differences between the different rainfall data sets could be identified, for catchment Pionierbrücke in WaSiM slight differences between method V1 and methods V2 and V3 differences regarding the seasonal extreme values can be identified. For both, V2 and V3, subsequent steps after the rainfall disaggregation were applied to implement spatial consistence by simultaneous rainfall
occurrence at different rain gauges. This affects the simulated runoff at least for instantaneous peak flows in the summer and winter period. However, the number of subcatchments in HBV and by that the spatial resolution of the rainfall-runoff model can be increased, which is assumed to lead to more diverse results between V1, V2 and V3, similar as resulting from WaSiM.

For Pionierbrücke, as a fast-reacting, mountainous catchment, the absolute differences for the seasonal extreme flows
resulting from V1 or the data sets V2 and V3 for a flood with a return period of 50 years are approx. 5-8 m³/s during both the calibration and validation periods (see Fig. 16 and 17) using WaSiM. For the other two catchments Reckershausen and Tetendorf, the difference is expected to be smaller since both catchments are larger and cover a less steep area. Thus, no additional simulations with WaSiM have been carried out for these two catchments. In this context it should be mentioned, that WaSiM is a much more complex rainfall-runoff model than HBV with a high demand on meteorological input time
series (e.g. precipitation, temperature, humidity, wind speed and global radiation), which has to be available for the whole simulation period on an hourly time step.




## 6. Conclusions & Outlook

The aim of this investigation is to investigate the influence of different degrees of spatial consistence in disaggregated time series on simulated runoff statistics. The investigation is carried out for three meso-scale catchments in Lower Saxony, Germany, which differ in terms of their size, land use, soil and slope. For the disaggregation, a multiplicative, micro-canonical cascade model after Müller and Haberlandt (2015) is used. Since the disaggregation process is performed on a per station basis without taking into account neighboring stations, spatial consistence must be implemented afterwards. Here, a resampling algorithm based on Müller and Haberlandt (2015) is applied (named V2) as well as a more pragmatic approach where the same relative diurnal cycle is used for all stations on the one day (Haberlandt and Radtke, 2014, named V3). Nevertheless, investigations without subsequent steps to implement spatial consistence exist as well (Ding et al., 2016) and have been included in this investigation (named V1). The hypothesis tested in this study is that these different rainfall data sets lead to differences in the derived runoff statistics as well. The following conclusions can be drawn regarding the rainfall data set differences:

1. The resampling algorithm for the implementation of spatial consistence was applied on an hourly basis for the first time for distances smaller than 20 km for V2. The achieved values for the bivariate spatial rainfall characteristics are comparable to those from observations.

2. The bivariate spatial characteristics are underestimated by V1 and overestimated by V3 respectively.

3. While for the areal rainfall intensities, the exceedance curve leads to an expected order of V1<V2<V3, for the areal rainfall extremes, V2 and V3 result in similar values, both being higher than V1.

The generated rainfall data sets V1, V2 and V3 have been used as input for rainfall-runoff modeling to evaluate the influence of the above identified differences of rainfall characteristics. An application-based evaluation is important in terms of rainfall generation, since it provides a new perspective and hence new insights into the rainfall data (Müller and Haberlandt, 2016, Müller et al., 2017, Sikorska et al., 2017). For the simulations, the semi-distributed HBV model (Wallner et al., 2013) and the fully-distributed WaSiM model (Schulla, 1997, 2015) have been implemented. The essential findings are:

1. With the applied calibration process in HBV, a good representation of observed runoff statistics is possible for V1-V3 for the calibration period.

2. The data sets V1-V3 result in only small differences in the simulated runoff statistics using HBV. Differences do not increase whether a neutral parameter set without calibration is applied nor if the station density increases.

3. For peak flows in the summer and winter periods, slight differences resulting from V1 and both, V2 and V3, can be identified using WaSiM. V2 and V3 lead to comparable higher flood peaks than V1, which is consistent with extreme value analysis of areal rainfall for this catchment.

4. For the intra-annual cycle and the flow duration curve, no difference resulting from V1-V3 can be identified from either HBV nor WaSiM.



Ding et al. (2016) achieved a good representation of summer and winter peak flows with V1 as input rainfall data and HBV as rainfall-runoff model. Haberlandt and Radtke (2014) applied HEC-HMS as semi-distributed rainfall-runoff model with disaggregated and parallelized rainfall time series (V3) as input data. The continuously simulated runoff time series were analyzed regarding annual extreme flows, which could be reproduced well for all catchments. The findings of both

investigations can be confirmed by the actual study.

However, no differences resulting from V1, V2 and V3 regarding the summer and winter extremes are detectable for HBV. This is remarkable, since V1 and V2 and their influence on simulated discharge have been analyzed before for 5-minute time steps by Müller and Haberlandt (2016) in an urban hydrological context. In their investigation, the urban hydrological model SWMM (Rossmann, 2010) has been used, which enables a good representation of the hydrodynamic of the fast-responding

sewage system. Significant differences could be identified between the simulated runoff statistics resulting from V1 and V2 for their artificial sewage system.

On the other hand, WaSiM results in slight differences for seasonal extreme values for the investigated catchment Pionierbrücke which is in line with previous findings regarding the areal rainfall extreme values. This is presumably caused by the spatial resolution, which is in WaSiM with 150 m x 150 m for each raster cell, being much higher than in HBV, with

approx. 20 km² per subcatchment. The higher spatial resolution enables higher spatial rainfall diversity, which is intensified by the rainfall corrected for altitude. A numerical diffusion of the rainfall in space due to a too-coarse spatial resolution is thereby avoided. Semi-distributed rainfall-runoff models like HBV or HEC-HMS with a simple horizontal structure and hence a less complex model-structure (as in e.g. WaSiM) lead to numerical diffusion and hence to a "smudging" of the areal rainfall, resulting in less differences in runoff statistics. Other investigations raise the question if spatial rainfall patterns can

be transferred sufficiently into runoff with semi-distributed models and thus with a coarse spatial resolution (Krajeski et al., 1991, Obled et al., 1994, Bárdossy and Das, 2008).

However, the differences between the resulting seasonal peak flows simulated with WaSiM from V1, V2 and V3 are still small with approx. 5-8 m³/s (up to 15 %) for floods with return periods of 50 years. It should be noted that V1, V2 and V3 clearly differ regarding the investigated spatial bivariate characteristics of probability of occurrence, coefficient of

correlation, continuity ratio and the resulting areal rainfall intensities, especially regarding their extreme values. Hence, the hypothesis formulated before is rejected in this case study. Although several possible causes regarding the applied rainfall-runoff models (parameter calibration, rainfall station density, type and spatial resolution of rainfall-runoff model) have been analyzed, no final conclusion about the reason for the similar runoff statistic can be drawn. It is assumed that the damping behavior of the catchments leads to these small differences in runoff statistics.

These findings suggest that (i) simple model structures might compensate for deficiencies in spatial representativeness through parameterization and (ii) highly resolved hydrological models benefit from improved spatial modeling of rainfall.

Of course, the similarity of the simulated runoff statistics from V1, V2 and V3 is only valid for the investigated catchments. For catchments with other climatic or physiographic attributes, results can be different. Therefore, a systematic investigation of catchments with different hydrological behavior in climates and with different rainfall-runoff models would be necessary



(comparative hydrology) to identify catchments, for which the degree of spatial rainfall consistence matters. The actual study could be a starting point for that.

However, the main intention of the actual study was to analyze the impact of rainfall data sets with different degrees of spatial consistence on simulated runoff statistics. The application of the resampling algorithm (V2) is recommended for

5  spatial application of disaggregated rainfall data since this method leads to the best agreement with the observed spatial rainfall characteristics.

**Competing interests**

The authors declare that they have no conflict of interest.

10  **Acknowledgements**

The authors thank the former student Jennifer Ullrich for calibration of the simulated annealing parameters. Thanks also to Ross Pidoto for useful comments on an early draft of the manuscript. A special thank to Bastian Heinrich for technical support during the investigation. We are also thankful for the permission to use the data of the German National Weather Service. The publication of this article was funded by the Open Access fund of Leibniz Universität Hannover.





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
