# Peer review of "Rainfall disaggregation for hydrological modeling: Is there a need for spatial consistence?"

_Hydrology and Earth System Sciences, 2017_

## Referee Comment (RC1) · A.E. Sikorska (Referee) · 21 Dec 2017

This manuscript deals with the problem of disaggregating daily rainfall records into an hourly resolution and particularly assesses the added value of considering the spatial correlation between neighboring stations. The problem of rainfall disaggregation is particularly relevant for hydrological modelling when only daily data are available but high resolution simulations are still required as for flood forecasting or flood predictions in case of fast reacting catchments (with the concentration time smaller than a day). Thus, the manuscript is certainly of a broader interest. Generally, it is also well written. However there are several issues that need more explanations and thus I recommend a major revision.

**Major comments:**

1. One of my major comments is related to the hypothesis tested by the authors. The authors assume that differences apparent between differently disaggregated rainfall series (V1, V2, V3) will also be present in runoff simulations with a hydrological model. For testing this hypothesis, they use a bucket-type model (HBV). From their results, the authors did not observe any significant difference between these three different rainfall series when fed into the model. I think this is not surprising because this type of model, due to its rather simple structure, may smooth slight differences between different time series present at instant steps, as it reacts to the cumulative rainfall sums over the event rather than to its small variation in time. Despite that, I would expect that you could still see some differences if you analyze instantaneous flows (e.g. event peaks). Yet, if you look only at cumulative statistics of runoff such as monthly average discharges or flow duration curve, you most likely cannot see any differences because these statistics are derived from averaged runoff values. Consequently, these differences could indeed be minute. The only visible effect could be expected on summer and winter extremes. However, these extremes most likely occur in your catchments due to large (and most likely long lasting) rainfall events, for which an exact rainfall distribution within a day is less important. This may explain why you do not observe any differences in these statistics neither. I think these issues should be at least discussed in the manuscript, and particularly the choice of the runoff statistics for the method evaluation.

2. The authors use throughout the manuscript terms: recording and non-recording stations. It is however never explained what they mean with that and I assume this is not a generally used term and thus should be explained as it is significant for this manuscript. It appears that by recording stations they mean stations with hourly records and by non-recording - stations with daily records only. To my understanding, daily stations could also be assumed as recording. Consider

using different terms or provide an explanation to the terms used.

3. Not all important details regarding the calibration of the hydrological model is given. Particularly, the fact that the model is calibrated independently with three different disaggregated datasets appears only in the discussion. These independent calibrations obviously lead to different parameter sets which are then used for simulating runoff. As these calibrated parameter sets compensate for possible errors in the model structure and in rainfall data, these errors are propagated on the simulated runoff (and computed statistics). This makes a direct comparison of runoff simulated with these three different time series difficult. Although the authors are aware of that, in my opinion, it would make more sense to use the model with the same set up. In this way, you could focus only on the effect of different rainfall time series and minimize the possible effect of parameter and model errors. Indeed, it could be worth a try to use only parameter sets derived from one calibration, e.g., with the V1 data set, and use it for both other disaggregated sets, i.e., V2 and V3 and in this way assess the gained effect of introducing the spatial consistence between stations into the set V1.

4. The figure 1 and the Table 1 suggest that each of three studied catchment is divided into smaller sub-catchments. Do you actually use these sub-catchments for hydrological modelling or do you model the catchment as one unit? I expect that the spatial representation of rainfall may play a role when using sub-catchments instead of the entire catchment.

5. The disaggregation scheme (p. 8): how exactly do you decide which time step is considered to be wet and which as dry? Also, is the same disaggregation scheme used for all three catchments (i.e., the same scheme of distributing daily totals into hourly intervals) or is that adapted for each catchment independently? In addition, the authors write in lines 16-17 p. 8 that parameters of the disaggregation (which exactly?) are extracted directly from the observed high resolution

data, which data do you mean exactly (the most recent hourly data)?

6. The disaggregation V3 uses the station with the highest values per day for deciding on an exact disaggregation way. Can you somehow verify that, i.e. how good it works for other stations? or could you justify this choice?

**Minor comments:**

1. P. 5 l. 12-13: change the sentence into: An overview of rain gauges used in this study is given in Fig. 1 while their measuring periods in Tab. 2.

2. Table 2; it could be a good idea to add the intervals of rainfall recording.

3. Use "rainfall-runoff model" instead of "rainfall-runoff-model" throughout the manuscript.

4. P. 6 l. 12-13: could you give a reference for the finding regarding the non-sensitivity to potential evapotranspiration?

5. P. 6. L. 7-9: is temperature data corrected for the elevation and if yes how exactly?

6. p. 7 l. 9-10: do you mean here "hourly" observed time series? From the table 3 it appears that some records are available from much longer period.

7. p. 7. L. 13-14: it is not clear which data sources were used to extract the maxima over half of year (hourly, daily, monthly)?

8. Use terms "section" and "subsection" instead of "chapter" and "subchapter".

9. The paragraph in lines 18-21 on p. 7 could also be removed.

10. p. 10, l. 15: how many different realizations of the disaggregated time series did you use for these simulations?

11. P. 10. L. 26, R1 is not explained before.

12. Table 5. The values for the HBV parameters: $k1$, $k1$, $k2$ and $kperc$ are given in days. If the model is run at an hourly time interval, should not these parameters be expressed in hours?

13. Fig, 5 and next: one realization from how many?

---

## Referee Comment (RC2) · N Peleg (Referee) · 5 Feb 2018

In their paper, the authors explored runoff response to hourly rainfall series with different degrees of spatial consistence. Daily rainfall series were disaggregated using a multiplicative random cascade method to generate 3 rainfall products – one without spatial consistency, and two others with a different level of spatial consistency. The question of the need for spatial consistence of rainfall disaggregation for hydrological modeling is interesting and relevant for the readers of HESS. My concerns are mostly minor, but I do have one major concern: I am not convinced that the HBV model is the right model to use for this experiment. Firstly, the model parameterization can overcome the differences in distributed rainfall products (as is also mentioned by the authors). Secondly, the area of the sub-catchments is very large (>20 km$^2$), so the rainfall spatial variability is essentially not introduced into the model. What was the reasoning in choosing HBV model? The studied catchments are rather small and distributed hydrological models (as WaSiM) could be easily applied. Other than that, the text requires some further editing. There is a disproportion between the length of the text and the number of figures and tables. I suggest reducing the length of the manuscript (the text is very repetitive) and have some of the figures/tables as supplementary information. Moreover, the terminology is inaccurate in some places. My recommendations for the text editing, along with some minor comments, are listed below. Overall I think the numerical experiment suggested by the authors is sound, and that the hydrology community will benefit from the paper. If my discussion below seems critical, it is only because I want to improve the final manuscript.

[Page Lines]

[introduction] I am missing some discussion about the importance of rainfall spatial variability to the runoff in general. The focus is mainly on the number of rain-gauges needed, but the readers will benefit from the understanding that it is important to capture (by dense rain-gauge networks, remote sensing or modeling) the rainfall spatial pattern right, as using a single rain-gauge or a single time series of areal rainfall the simulated runoff is likely to be over-(under)estimated. I can think of several papers that discuss this point: Gires et al. (2012, JoH); Gires et al. (2013, UWJ); Paschalis et al. (2014; JoH); Ochoa-Rodriguez et al. (2015, JoH); Peleg et al. (2017, HESS).

[2 11] "non-recording stations" and "recording stations" – I would adopt a simpler terminology, e.g. "hourly stations" and "daily stations" (or hourly-recording stations).

[2 12] "time series from non-recording stations can then be disaggregated" – same here, I would simplify the terminology used. Consider revising to "daily time series can be disaggregated to hourly…"

[2 15] "over other rainfall generators" – I suggest to replace this with "over other disaggregation methods" or similar. Rainfall generators preserve the statistics of a rainfall series but often not used to disaggregate a given time series while preserving the rainfall amount at the coarse scale.

[2 17] "higher" – finer.

[2 37] "three bivariate rainfall characteristics". Why 'bivariate'? You have a single variable, if only rainfall is explored. I would change the terminology to "three spatial rainfall characteristics" or "three spatial rainfall indices".

[3 8] "investigations" replace with "studies".

[3 12] "rainfall data sets" – they all emerge from the same data set, consider replacing with "rainfall products" or similar to distinguish from the original time series.

[3 20] "by amongst others" – change to "by others, as".

[3 22] "Runoff statistics have no connection to time" – please revise this sentence, runoff statistics are time dependent, e.g. statistics of runoff diurnal cycle.

[3 25] "to take into account different genesis" – not a clear sentence.

[3 27] "investigation area" – I believe "study area" is a more common phrase to use.

[3 27] replace "chapter" with "section".

[Fig. 1] What are "p-stations" and "gauges" stand for? I guess p-stations are rain-gauges and gauges stand for discharge-gauges. Please correct the legend accordingly.

[Table 2] The names of the gauges are not important for the readers (and are not labeled in Fig. 1 or 2). They can be removed to shorten the length of the table.

[6 12] "has been shown not to be that sensitive as model input" – a reference is needed here.

[7 6] "as per" – replace with "as in".

[7 11] Please define "monthly extreme values". Do you mean hourly extremes on monthly basis? If not, than I would expect monthly extreme values time series to start with the daily discharge time series.

[7 25] "The most important input for rainfall-runoff models are long and high-resolution rainfall time series from a dense rain gauge network" – This sentence is more suitable to the introduction section and it needs to be supported with a reference. High-resolution –

do you mean temporal and spatial? If so, also weather radars and, if catchments are large enough, satellite rainfall data can be used.

[7 25] to [8 3] This paragraph is somehow a repetitive to what was already stated in the introduction. It can be removed from the text.

[8 15] to [9 6] I think this part can be also removed as it is described (in details) in Müller and Haberlandt (2015). Unless there is some important information here that is later discussed. The disaggregation scheme is well illustrated (Fig. 3) and explained in the preceding paragraph.

[9 8] "b) Bivariate characteristics" – replace with "Rainfall spatial characteristics indices" or similar.

[9 9] to [9 13] "The disaggregation of single time series is carried out without taking into account time series of surrounding stations. For each time series the cascade model distributes the wet time steps randomly during a wet day due to its disaggregation scheme. Hence, spatial consistence of rainfall is underestimated after the disaggregation. Spatial consistence is defined in this investigation by bivariate spatial rainfall characteristics, the namely probability of occurrence, Pearson's coefficient of correlation, and the continuity ratio (Wilks, 1998)" – This all is a repetitive of what was already mention in the introduction. Be concise. I would replace the first paragraph of this sub-section with a single sentence, e.g. "The rainfall spatial characteristics are following the ones used by Haberlandt (2008) and are briefly described in the following".

[Eq. 1] "Z" – stands for rainfall intensity?

[Eq. 2] The "x" in the denominator should be deleted. It reads like a variable.

[Eq. 1 and 2] Consider removing them. I think that the text to describe Eq. 1 is sufficient for the readers to understand the rainfall occurrence score and Pearson's coefficient of correlation (Eq. 2) is quite well known.

[9 29] "(see Fig. 4)" – I don't see it.

[10 6] to [10 27] I found the description of V2 to be very long; I am not sure if the readers needs all this information about the method. I recommend to shorten this part (and removing Eq. 4). The concept of V2 is already explained at the introduction. Are there any modification from what is presented in Müller and Haberlandt (2015)?

[10 28] to [10 33] The part describing V3 is concise and well written, but again – there are many repeats to what was already written in the previous sections.

[Fig. 4 and Table 5] Can be moved to the Supplementary Information (SI).

[11 23] to [11 26] A repetitive.

[12 1] Please define the periods for summer and winter.

[12 10] FDC should be calculated on both hourly and daily scales! Important information can be obtained from exploring both scales. What is the point in disaggregating the rainfall to hourly scale and examining it on daily scale?

[Eq. 5] Remove the "max".

[Eq. 6] Can be moved to the SI.

[Eq. 7] Remove "min". What is the logic behind the weights?

[Table 5] Move to SI.

[Section 3.2a] I am missing the information about how the HBV model is "distributed" in space. Are the catchments represented as one unit or many? If many, I would like to see a figure with how the units are distributed in space (following the sub-catchments illustrated in Fig. 1 and 2?). This is a critical point as you later discuss the spatial representation of rainfall over the catchments, but it is not clear how the model relate to rainfall in space.

[14 1] "The spatial resolution of WaSiM applications covers several scales ranging from tens of meters to a few kilometers" – but what is used here? For example, what was the spatial resolution of the modeled rainfall?

[14 22] The calibration period is quite short, isn't it?

[15 2] to [15 13] and [Table 7] I do not see the need in repeating the results reported in Müller and Haberlandt (2015) here. It can be replaced with a one line sentence indicating the method advantages and limitations, but this should be anyhow done prior to the result section (i.e. in the methods section).

[Fig. 5] Where are all the dots (other rain-gauges) coming from? Is there any reason to present the different scores for a distance of 250 km? I recommend limiting the distance to 25 or 50 km, to agree with the catchment size. Please give some information about the fitting that are presented. If the fits are not discussed, than the lines can be removed. It will be useful to have the same figure for the other catchments in the SI.

[16 15] "areal rainfall intensity" – please define how it was calculated. A simple arithmetic mean?

[Fig. 6] Can also be moved to the SI, to reduce the number of figures in the paper.

[17 14] "runoff".

[17 15] Why Weibull? Is the fit good for all rainfall products (V1 to V3)? What is the length of the rainfall series used to generate Fig. 7?

[Fig. 7] What is the size of sub-catchment 2? From Figure 1 I would estimate around 20 km$^2$. If this is the case, I would argue that V2 and V3 are likely overestimating the extreme rainfall intensities. For example, ~38 mm h$^{-1}$ for a 10-year return period over a ~20 km$^2$ sounds quite a lot for me. It can be reasonable for a measurement from a single rain-gauge, but as we are looking at areal rainfall I would expect much lower values – even in the range of V1 (as the extreme rainfall intensity is expected to be smaller for the same return period when shifting from a point scale to a larger areal scale, e.g. our recent study in JoH [Peleg et al., 2018] and many others). I would suggest to compare the resulted areal rainfall to an observed extreme rainfall from a single-gauge, even the comparison will be areal to point, just to get a sense of the differences between the scales.

[19 4] "flood quantiles are shown for a return period of 100 years" – It doesn't make sense as the observed period is much shorter. I would focus on 50 y return period to reduce the uncertainties.

[Fig. 12] For which catchment? Or is it for the entire region? Same comments as of Fig. 5 above.

[25 5] to [25 10] More suitable to be in the Introduction section.

[27 13] "It can be summarized, that the number of rain gauges has only a minor, but no systematic influence on runoff statistics for the catchments used in this investigation" – but likely not because of the number of rain-gauges in a catchment but because of the hydrological model that was used! I would like to see the same analysis using a fully distributed hydrological model that can account for spatial rainfall variability at the sub-catchment scales.

[27 14] A repetitive.

[Table 12 and 13] SI.

[31 8] to [31 15] There are some repetitive here is well.

[32 9] "the IDW algorithm with an altitudinal rainfall adjustment, which was carried out by a linear regression model" – The IDW is likely to smooth the rainfall in space, thus reducing the spatial rainfall variability and the variability in flow.

[32 10] But FDC compare daily discharges, right? I guess that at hourly scale the differences are clearer.

[38 6] to [38 11] Part of the reason is because the sub-catchments sets in SWMM model are often much finer than the 20-km set by HBV in this study. When exploring the hydrology response using small sub-catchments with SWMM the effect of the distributed rainfall in space are evident on the hydrological flows (see for example the study by Peleg et al. 2017, HESS). I have reasons to believe that if HBV model was set to have many more small sub-catchments for this study, the results of the differences between V1, V2 and V3 would look different. That rise the question of the suitability of HBV model with the current setting of a few large sub-catchment to explore the sensitivity of the hydrological response to different rainfall spatial characteristics.

[Conclusion section] The conclusion part is a mix of discussion, summary and conclusions. Consider revising it to make the outcome of the experiment clearer to the readers.

---

## Author Comment (AC1) · 5 Mar 2018

This manuscript deals with the problem of disaggregating daily rainfall records into an hourly resolution and particularly assesses the added value of considering the spatial correlation between neighboring stations. The problem of rainfall disaggregation is particularly relevant for hydrological modelling when only daily data are available but high resolution simulations are still required as for flood forecasting or flood predictions in case of fast reacting catchments (with the concentration time smaller than a day). Thus, the manuscript is certainly of a broader interest. Generally, it is also well written. However there are several issues that need more explanations and thus I recommend a major revision.

The authors thank the reviewer for her comments and suggestions. All replies are indicated with page and lines, based on the track-change document.

**Major comments:**
1. One of my major comments is related to the hypothesis tested by the authors. The authors assume that differences apparent between differently disaggregated rainfall series (V1, V2, V3) will also be present in runoff simulations with a hydrological model. For testing this hypothesis, they use a bucket-type model (HBV). From their results, the authors did not observe any significant difference between these three different rainfall series when fed into the model. I think this is not surprising because this type of model, due to its rather simple structure, may smooth slight differences between different time series present at instant steps, as it reacts to the cumulative rainfall sums over the event rather than to its small variation in time. Despite that, I would expect that you could still see some differences if you analyze instantaneous flows (e.g. event peaks). Yet, if you look only at cumulative statistics of runoff such as monthly average discharges or flow duration curve, you most likely cannot see any differences because these statistics are derived from averaged runoff values. Consequently, these differences could indeed be minute. The only visible effect could be expected on summer and winter extremes. However, these extremes most likely occur in your catchments due to large (and most likely long lasting) rainfall events, for which an exact rainfall distribution within a day is less important. This may explain why you do not observe any differences in these statistics neither. I think these issues should be at least discussed in the manuscript, and particularly the choice of the runoff statistics for the method evaluation.

We thank reviewer 1 for raising her concerns about the choice of the model type. We try to discuss all concerns in the order of her comment. Reviewer 1 is worried about only slight differences between the different rainfall products V1, V2 and V3. However, as shown in Fig. 7 for areal rainfall of one subcatchment, the extreme values differ strongly (5-12 mm) between the different rainfall products for return periods from 2-50 years. Despite that, more frequent rainfall intensities also differ as indicated by Fig. 5 and Fig. 6. So there are differences between the rainfall products.

Also, the reviewer questions the type of model for the actual investigation. We have added a discussion on that in the manuscript (p13 15-21) and want to point out, that the catchments were divided into subcatchments (see also our reply to comment 4 of reviewer 1).

Nevertheless, we agree with reviewer 1, the runoff statistics FDC and Qmon on a daily basis are not suitable to show differences between the rainfall products. However, this was never the intention. Both runoff statistics have been applied only to achieve an overall plausible runoff behaviour for the continuous simulations. We have added a brief discussion on p15 5-9:

"FDC and Q-mon are used to represent the more frequent discharge values. Q-mon accounts for the temporal dependency on the inter-annual variation of the discharge. As mentioned before, the analyzes of FDC and Q-mon allows no direct validation of the rainfall products, but enables an overall plausible simulation of rainfall-runoff processes."

We thank reviewer 1 for the useful suggestion with the possibility of long-lasting events as main reasons for extreme runoff values. Therefore, we have distinguished between summer and winter extremes, so that e.g. the convective events in the fast responding catchment Pionierbrücke and

long-lasting events in the winter period are taken into account. We mention the season-dependent genesis and its influence on runoff extremes on p3 34-p4 1:
"to take into account e.g. summer and winter floods with their different genesis and resulting runoff behaviour."

2. The authors use throughout the manuscript terms: recording and non-recording stations. It is however never explained what they mean with that and I assume this is not a generally used term and thus should be explained as it is significant for this manuscript. It appears that by recording stations they mean stations with hourly records and by non-recording - stations with daily records only. To my understanding, daily stations could also be assumed as recording. Consider using different terms or provide an explanation to the terms used.
We thank the reviewer for this hint. We have rephrased both terms to hourly (former: recording) and daily (former: non-recording) stations and believe, these terms are more intuitive for the reader.

3. Not all important details regarding the calibration of the hydrological model is given. Particularly, the fact that the model is calibrated independently with three different disaggregated datasets appears only in the discussion. These independent calibrations obviously lead to different parameter sets which are then used for simulating runoff. As these calibrated parameter sets compensate for possible errors in the model structure and in rainfall data, these errors are propagated on the simulated runoff (and computed statistics). This makes a direct comparison of runoff simulated with these three different time series difficult. Although the authors are aware of that, in my opinion, it would make more sense to use the model with the same set up. In this way, you could focus only on the effect of different rainfall time series and minimize the possible effect of parameter and model errors. Indeed, it could be worth a try to use only parameter sets derived from one calibration, e.g., with the V1 data set, and use it for both other disaggregated sets, i.e., V2 and V3 and in this way assess the gained effect of introducing the spatial consistence between stations into the set V1.
Again, we want to reply to all points/suggestions in the order of appearance. We added the information about the separate calibration for each rainfall product in the method section (p14 18). Also, we are aware of the possible compensation of rainfall product differences by the model parameters. This is the reason for investigations "c1) HBV-simulation results without calibration using three rain gauges as input" and "c2) WaSiM- simulation results without calibration using three rain gauges as input". For both, c1 and c2, no calibration was carried out. A neutral parameter set was applied to avoid (dis-) advantages for one of the rainfall products/biased results. Nevertheless, no differences with HBV and only slight differences with WaSiM could be identified.

4. The figure 1 and the Table 1 suggest that each of three studied catchments is divided into smaller sub-catchments. Do you actually use these sub-catchments for hydrological modelling or do you model the catchment as one unit? I expect that the spatial representation of rainfall may play a role when using sub-catchments instead of the entire catchment.
We agree with the argument of the reviewer, that subcatchments are important for the investigation of the spatial consistence. The discretization shown in Fig. 1 in several subcatchments (with approx. 20 km²) is applied for the investigation.

5. The disaggregation scheme (p. 8): how exactly do you decide which time step is considered to be wet and which as dry? Also, is the same disaggregation scheme used for all three catchments (i.e., the same scheme of distributing daily totals into hourly intervals) or is that adapted for each catchment independently? In addition, the authors write in lines 16-17 p. 8 that parameters of the disaggregation (which exactly?) are extracted directly from the observed high resolution data, which data do you mean exactly (the most recent hourly data)?
The brief explanation of the cascade model leads to open questions by reviewer 1. However, in accordance with our reply to reviewer 2 we have even shortened the description in the manuscript and strongly refer the interested reader to the original manuscript by Müller and Haberlandt (2015). However, we want to answer the questions also in our reply. The decision about the wetness-state of each time step is made randomly, based on probabilities estimated from observed time series.

Hence, a second run of the disaggregation leads to a different time series. This is the reason why 10 different realizations have been used as input for the rainfall-runoff simulations. So the scheme in Fig. 3 shows only one possible realization for a single day. The parameters are estimated from the nearest hourly stations with a minimum record length of 7 years. This could be the most recently hourly time series or from a station installed from e.g. 1980-2002 For a description of the cascade model parameters the reviewer is kindly referred to Müller and Haberlandt (2015).

6. The disaggregation V3 uses the station with the highest values per day for deciding on an exact disaggregation way. Can you somehow verify that, i.e. how good it works for other stations? or could you justify this choice?

Reviewer 1 is confused by V3. V3 is not a new disaggregation method, only an alternative to V2 to implement spatial consistence into the disaggregated time series of V1. We have tried to clarify this by adding an additional sentence (12 27): " and is also based on the already disaggregated time series of V1."

**Minor comments:**
1. P. 5 l. 12-13: change the sentence into: An overview of rain gauges used in this study is given in Fig. 1 while their measuring periods in Tab. 2.
Thanks for the suggestion, we have rephrased the sentence.
2. Table 2; it could be a good idea to add the intervals of rainfall recording.
We thank the reviewer for the hint. We have changed the table and now the temporal resolution of the stations is included.
3. Use "rainfall-runoff model" instead of "rainfall-runoff-model" throughout the manuscript.
Thanks for the hint, we have rephrased it throughout the manuscript.
4. P. 6 l. 12-13: could you give a reference for the finding regarding the nonsensitivity to potential evapotranspiration?
The analyzes was carried out in a pre-study by a student (Herzog, 2013) and is hence not citable in a scientific journal. However, the HBV-IWW model was analyzed regarding its sensitivity to input time series of rainfall, temperature and potential evapotranspiration. The model was not sensitive to the latter one.
5. P. 6. L. 7-9: is temperature data corrected for the elevation and if yes how exactly?
The temperature data has not been corrected. Based on station data, an interpolation was carried out for the study area using External Drift Kriging. The additional information used is elevation.
6. p. 7 l. 9-10: do you mean here "hourly" observed time series? From the table 3 it appears that some records are available from much longer period.
We thank reviewer 1 for pointing out the missing information. Indeed, it should be "observed hourly time series". We have rephrased it.
7. p. 7. L. 13-14: it is not clear which data sources were used to extract the maxima over half of year (hourly, daily, monthly)?
We thank reviewer 1 for pointing this out. We added the data sources to avoid misinterpretations (p8 23-224).
8. Use terms "section" and "subsection" instead of "chapter" and "subchapter".
Thanks for the hint, we have rephrased it throughout the manuscript.
9. The paragraph in lines 18-21 on p. 7 could also be removed.
We thank reviewer 1 for the suggestion, but due to the length of section 3 we think that a brief overview at the beginning enables a better understanding of the investigation.
10. p. 10, l. 15: how many different realizations of the disaggregated time series did you use for these simulations?
We are thankful for pointing out the missing information. We have used 10 realizations for each rainfall product and added the information (p14 18-19).
11. P. 10. L. 26, R1 is not explained before.
We have removed this paragraph.

12. Table 5. The values for the HBV parameters: k1, k1, k2 and kperc are given in days. If the model is run at an hourly time interval, should not these parameters be expressed in hours?

We thank the reviewer for his concern about the units. As it can be identified for k0, not only integer values are possible (minimum for k0=0.25 d = 6 h). The unit [d] was only chosen here for a better understanding of the value, e.g. k2=500 d is more suitable/easier to follow in comparison to k2= 12000 h.

13. Fig, 5 and next: one realization from how many?

10 realizations for each rainfall product, please see our answer to minor comment 10.

References:

Müller, H. und Haberlandt, U. (2015). „Temporal Rainfall Disaggregation with a Cascade Model: From Single-Station Disaggregation to Spatial Rainfall". *Journal of Hydrologic Engineering* 20 (11), p. 04015026.

Herzog, Y. (2013): "Sensitivität eines hydrologischen Modells auf Veränderungen in den Klimavariablen Niederschlag, Temperatur und potentielle Verdunstung". *Student thesis* at the Institute of Water Resources Management, Hydrology and Agricultural Hydraulic Engineering, Faculty of Civil Engineering and Geodesy, Leibniz Universität Hannover (in German).

---

## Author Comment (AC2) · 5 Mar 2018

**Comments Reviewer 2**

In their paper, the authors explored runoff response to hourly rainfall series with different degrees of spatial consistence. Daily rainfall series were disaggregated using a multiplicative random cascade method to generate 3 rainfall products – one without spatial consistency, and two others with a different level of spatial consistency. The question of the need for spatial consistence of rainfall disaggregation for hydrological modeling is interesting and relevant for the readers of HESS. My concerns are mostly minor, but I do have one major concern: I am not convinced that the HBV model is the right model to use for this experiment. Firstly, the model parameterization can overcome the differences in distributed rainfall products (as is also mentioned by the authors). Secondly, the area of the sub-catchments is very large (>20 $km_2$), so the rainfall spatial variability is essentially not introduced into the model. What was the reasoning in choosing HBV model? The studied catchments are rather small and distributed hydrological models (as WaSiM) could be easily applied. Other than that, the text requires some further editing. There is a disproportion between the length of the text and the number of figures and tables. I suggest reducing the length of the manuscript (the text is very repetitive) and have some of the figures/tables as supplementary information. Moreover, the terminology is inaccurate in some places. My recommendations for the text editing, along with some minor comments, are listed below. Overall I think the numerical experiment suggested by the authors is sound, and that the hydrology community will benefit from the paper. If my discussion below seems critical, it is only because I want to improve the final manuscript.

The authors thank the reviewer for his comments and suggestions. All replies are indicated with page and lines, based on the track-change document.

[Page Lines]
[introduction] I am missing some discussion about the importance of rainfall spatial variability to the runoff in general. The focus is mainly on the number of rain-gauges needed, but the readers will benefit from the understanding that it is important to capture (by dense rain-gauge networks, remote sensing or modeling) the rainfall spatial pattern right, as using a single rain-gauge or a single time series of areal rainfall the simulated runoff is likely to be over-(under)estimated. I can think of several papers that discuss this point: Gires et al. (2012, JoH); Gires et al. (2013, UWJ); Paschalis et al. (2014; JoH); Ochoa-Rodriguez et al. (2015, JoH); Peleg et al. (2017, HESS).
We thank the reviewer for the useful references and have implemented a selection of the references in the manuscript (p2 10-13).
[2 11] "non-recording stations" and "recording stations" – I would adopt a simpler terminology, e.g. "hourly stations" and "daily stations" (or hourly-recording stations).
Indeed, a simpler terminology improves the understanding. We have changed the names to "hourly stations" and "daily stations".
[2 12] "time series from non-recording stations can then be disaggregated" – same here, I would simplify the terminology used. Consider revising to "daily time series can be disaggregated to hourly…"
In accordance with the former comment we have rephrased this part.
[2 15] "over other rainfall generators" – I suggest to replace this with "over other disaggregation methods" or similar. Rainfall generators preserve the statistics of a rainfall series but often not used to disaggregate a given time series while preserving the rainfall amount at the coarse scale.
We thank the reviewer for pointing out the misleading sentence. The advantage of the cascade model mentioned here is the temporal reference to observations, and hence, the facilitation of their statistics down to finer temporal resolutions. On the contrary, rainfall generators create an

hourly time series "out of nothing", which is a more difficult task (although they have other advantages). We have tried to clarify this point by rephrasing this sentence (p2 18-23).

[2 17] "higher" – finer.

Thanks for the hint, we have rephrased it.

[2 37] "three bivariate rainfall characteristics". Why 'bivariate'? You have a single variable, if only rainfall is explored. I would change the terminology to "three spatial rainfall characteristics" or "three spatial rainfall indices".

The reviewer is concerned about a wrong terminology. The rainfall characteristics are "bivariate", since they result from the comparison of exactly two time series. Each comparison leads to one point of the e.g. observation cloud in Fig. 5. In a multivariate analysis, all time series are taken into account with only one resulting, final point/value. Hereby, the distance-dependency gets lost. The term "bivariate" is common for these characteristics from the point of the authors view (Wilks (1998), Haberlandt et al. (2008), Müller und Haberlandt (2015, 2018)) and is hence kept in the manuscript.

[3 8] "investigations" replace with "studies".

Thanks for the hint, we have rephrased it.

[3 12] "rainfall data sets" – they all emerge from the same data set, consider replacing with "rainfall products" or similar to distinguish from the original time series.

Thanks for the hint, we have replaced it by "rainfall products" throughout the whole manuscript.

[3 20] "by amongst others" – change to "by others, as".

Thanks for the hint, we have rephrased it.

[3 22] "Runoff statistics have no connection to time" – please revise this sentence, runoff statistics are time dependent, e.g. statistics of runoff diurnal cycle.

The reviewer is confused by the terminology, "no connection to time". Indeed, runoff statistics have always a connection to time, e.g. the period or season of runoff time series used for analyzes, or, as the reviewer points out, single hours of a day to achieve a representative diurnal cycle. However, the all these statistics have no connection to a concrete time step/day/month – so it is impossible conclude from a statistic value (e.g. $HQ_{100}$ or average discharge in the 5$^{th}$ hour of a day) to its day of occurrence (it it has ever occurred!). We have tried to solve the issue be rephrasing the sentence to "Runoff statistics are time-independent" (p3 31).

[3 25] "to take into account different genesis" – not a clear sentence.

Thanks for the hint, we have rephrased it to "to take into account summer and winter floods with their different genesis and resulting runoff behavior" (p3 34-p4 2).

[3 27] "investigation area" – I believe "study area" is a more common phrase to use.

Thanks for the hint, we have rephrased it.

[3 27] replace "chapter" with "section".

Thanks for the hint, we have rephrased it throughout the whole manuscript.

[Fig. 1] What are "p-stations" and "gauges" stand for? I guess p-stations are rain-gauges and gauges stand for discharge-gauges. Please correct the legend accordingly.

We have changed it according to the reviewer's suggestion.

[Table 2] The names of the gauges are not important for the readers (and are not labeled in Fig. 1 or 2). They can be removed to shorten the length of the table.

We thank the reviewer for the useful suggestion and have changed the table accordingly.

[6 12] "has been shown not to be that sensitive as model input" – a reference is needed here.

The analyzes was carried out in a pre-study by a student (Herzog, 2013) and is hence not citable in a scientific journal. However, the HBV-IWW model was analyzed regarding its sensitivity to input time series of rainfall, temperature and potential evapotranspiration. The model was not sensitive to the latter one.

[7 6] "as per" – replace with "as in".

Thanks for the hint, we have rephrased it.

[7 11] Please define "monthly extreme values". Do you mean hourly extremes on monthly basis? If not, than I would expect monthly extreme values time series to start with the daily discharge time series.

Thanks for the hint, we have rephrased it to the reviewers suggestion "hourly extreme values on a monthly basis" (p8 20-21).

[7 25] "The most important input for rainfall-runoff models are long and high-resolution rainfall time series from a dense rain gauge network" – This sentence is more suitable to the introduction section and it needs to be supported with a reference. High-resolution – do you mean temporal and spatial? If so, also weather radars and, if catchments are large enough, satellite rainfall data can be used.

The reviewer is right with his argument, that this sentence fits better to the introduction part. Indeed, the information has already been given in section 1 (with references). Hence, we have deleted the sentences here.

[7 25] to [8 3] This paragraph is somehow a repetitive to what was already stated in the introduction. It can be removed from the text.

Please see our reply to the former comment.

[8 15] to [9 6] I think this part can be also removed as it is described (in details) in Müller and Haberlandt (2015). Unless there is some important information here that is later discussed. The disaggregation scheme is well illustrated (Fig. 3) and explained in the preceding paragraph.

We thank the reviewer for the useful suggestion and have deleted the paragraph.

[9 8] "b) Bivariate characteristics" – replace with "Rainfall spatial characteristics indices" or similar.

The reviewer is referred to our reply to comment [2 37].

[9 9] to [9 13] "The disaggregation of single time series is carried out without taking into account time series of surrounding stations. For each time series the cascade model distributes the wet time steps randomly during a wet day due to its disaggregation scheme. Hence, spatial consistence of rainfall is underestimated after the disaggregation. Spatial consistence is defined in this investigation by bivariate spatial rainfall characteristics, the namely probability of occurrence, Pearson's coefficient of correlation, and the continuity ratio (Wilks, 1998)" – This all is a repetitive of what was already mention in the introduction. Be concise. I would replace the first paragraph of this sub-section with a single sentence, e.g. "The rainfall spatial characteristics are following the ones used by Haberlandt (2008) and are briefly described in the following".

We thank the reviewer for his suggestion. We have replaced the introduction paragraph (p11 2-3).

[Eq. 1] "Z" – stands for rainfall intensity?

Yes, we have added the missing information.

[Eq. 2] The "x" in the denominator should be deleted. It reads like a variable.

We have replaced the misleading operation sign by a representative operator.

[Eq. 1 and 2] Consider removing them. I think that the text to describe Eq. 1 is sufficient for the readers to understand the rainfall occurrence score and Pearson's coefficient of correlation (Eq. 2) is quite well known.

We are thankful for the suggestion of the reviewer. However, the third characteristic (Eg.3) is rather known and should remain in the manuscript. To keep it consistent, we keep both equations. They are also essential for the validation of the spatial consistence.

[9 29] "(see Fig. 4)" – I don't see it.

We thank reviewer 2 for this hint. Indeed, the regression lines are in the former Fig. 5, now in Fig. 4.

[10 6] to [10 27] I found the description of V2 to be very long; I am not sure if the readers needs all this information about the method. I recommend to shorten this part (and removing Eq. 4).

The concept of V2 is already explained at the introduction. Are there any modification from what is presented in Müller and Haberlandt (2015)?

We thank reviewer 2 for the useful suggestion. We have shortened the paragraph.

[10 28] to [10 33] The part describing V3 is concise and well written, but again – there are many repeats to what was already written in the previous sections.

We thank reviewer 2 for the hint. We have shortened the text in the introduction section to avoid repeats.

[Fig. 4 and Table 5] Can be moved to the Supplementary Information (SI).

We are thankful for the reviewers suggestion and have moved both to the SI.

[11 23] to [11 26] A repetitive.

We agree and have shortened the paragraph.

[12 1] Please define the periods for summer and winter.

We have implemented the missing information (p14 14-15).

[12 10] FDC should be calculated on both hourly and daily scales! Important information can be obtained from exploring both scales. What is the point in disaggregating the rainfall to hourly scale and examining it on daily scale?

Reviewer 2 is right- an evaluation of the rainfall products on an hourly basis with FDC and Q-mon on a daily basis can be questioned. However, this is not the idea behind including both runoff criteria. Rainfall products are expected to influence the extreme values in summer and winter half year. FDC and Q-mon are taken into account additionally to represent the overall behavior of the rainfall-runoff processes to e.g. achieve realistic filling volumes for all storages before a severe rainfall event occurs.

We have rephrased the whole paragraph and added some explanations for clarification (p15 2-10).

[Eq. 5] Remove the "max".

We have removed it.

[Eq. 6] Can be moved to the SI.

We thank reviewer 2 for the suggestion to shorten the text. However, a movement of Eq. 6 to the SI would result in an implementation of the applied quantiles in the text. A complete neglecting of the applied quantiles in the manuscript would complicate the understanding of the calibration procedure. Hence, we prefer to leave Eq. 6 in the manuscript.

[Eq. 7] Remove "min". What is the logic behind the weights?

We have removed it.

[Table 5] Move to SI.

We are thankful for the reviewers suggestion and have moved it to the SI.

[Section 3.2a] I am missing the information about how the HBV model is "distributed" in space. Are the catchments represented as one unit or many? If many, I would like to see a figure with how the units are distributed in space (following the sub-catchments illustrated in Fig. 1 and 2?). This is a critical point as you later discuss the spatial representation of rainfall over the catchments, but it is not clear how the model relate to rainfall in space.

Reviewer 1 points out the missing information about the spatial discretization of the HBV-model. Indeed, the subcatchments shown in Fig. 1 and 2 represent the spatial discretization. We have implemented additional sentences pointing it out in the method section along with a paragraph regarding model selection, providing also some information with a focus on the choice of interpolation method and its reasons (p13 12-27).

[14 1] "The spatial resolution of WaSiM applications covers several scales ranging from tens of meters to a few kilometers" – but what is used here? For example, what was the spatial resolution of the modeled rainfall?

We thank reviewer 2 for pointing out the misplaced information. The spatial resolution of WaSiM is included in the discussion section (150 m x 150 m)). We have implemented it in the method section as well (p16 10-11).

[14 22] The calibration period is quite short, isn't it?

For the choice of calibration and validation period a classical split-sampling was applied. Depending on the total period available, the first half of the period was used for the calibration, the second one for the calibration. Indeed, for Tetendorf it is maybe critical with14 years in total. However, this is discussed in the results section (p28 12).

[15 2] to [15 13] and [Table 7] I do not see the need in repeating the results reported in Müller and Haberlandt (2015) here. It can be replaced with a one line sentence indicating the method advantages and limitations, but this should be anyhow done prior to the result section (i.e. in the methods section).

We agree with the reviewer's suggestion and have shortened the results summary and moved it to the methods section (p9 21-24).

[Fig. 5] Where are all the dots (other rain-gauges) coming from? Is there any reason to present the different scores for a distance of 250 km? I recommend limiting the distance to 25 or 50 km, to agree with the catchment size. Please give some information about the fitting that are presented. If the fits are not discussed, than the lines can be removed. It will be useful to have the same figure for the other catchments in the SI.

We thank the reviewer for pointing out the missing information. The black circles result from observations (details are explained in Müller and Haberlandt, 2015). We have added the information in the figure captions. The analyzes was carried out for the other two catchments as well and can be found in the SI. According the suggestion the x-axes have been limited to 40 km.

[16 15] "areal rainfall intensity" – please define how it was calculated. A simple arithmetic mean?

The description was included in the discussion section. We have moved it to the method section (p13 21-23).

[Fig. 6] Can also be moved to the SI, to reduce the number of figures in the paper.

We thank the reviewer for this suggestion, the figure has been moved to SI.

[17 14] "runoff".

We have corrected it.

[17 15] Why Weibull? Is the fit good for all rainfall products (V1 to V3)? What is the length of the rainfall series used to generate Fig. 7?

The reviewer is concerned about the goodness-of-fit of the Weibull-distribution function for all rainfall products. Here, only the Weibull-plotting position has been used, which needs no fitting. It combines only the rainfall intensity and its rank in a sorted population for comparative analyzes. The length of the time series is 53 years, we have added the information in the figure caption.

[Fig. 7] What is the size of sub-catchment 2? From Figure 1 I would estimate around 20 km$^2$. If this is the case, I would argue that V2 and V3 are likely overestimating the extreme rainfall intensities. For example, ~38 mm h$_{-1}$ for a 10-year return period over a ~20 km$^2$ sounds quite a lot for me. It can be reasonable for a measurement from a single rain-gauge, but as we are looking at areal rainfall I would expect much lower values – even in the range of V1 (as the extreme rainfall intensity is expected to be smaller for the same return period when shifting from a point scale to a larger areal scale, e.g. our recent study in JoH [Peleg et al., 2018] and many others). I would suggest to compare the resulted areal rainfall to an observed extreme rainfall from a single-gauge, even the comparison will be areal to point, just to get a sense of the differences between the scales.

Reviewer 2 points out several issues regarding differences between point and areal rainfall, which we try to answer in the following. First, the reviewer's estimation of the catchment size

with approx. 20 km² is right (we have added the missing information regarding spatial distribution in the HBV-model description (p 13 14)). A comparison of the estimated areal rainfall with "observations" is not possible, as the reviewer points out himself, since only one hourly station is available for this catchment. A comparison, as suggested, with rainfall from a single-gauge is also not possible, since the maximum record length of available hourly data is shorter (and a comparison of extreme values resulting from different periods with different lengths would not be representative) and the station is located outside the catchment. Both not possible comparisons show the need for the rainfall disaggregation.

However, we totally agree with the reviewer's point of decreasing rainfall intensity with increasing area size. This is most important for high-resolution rainfall (as in Peleg et al., 2018), but decreases with increasing time step length due to cumulative character of a single time step (cumulating rainfall i) over time and due to e.g. cloud movements ii) over space). However, simulations are carried out continuously and hence, also underestimations occur in the actual setup (Müller and Haberlandt, 2018), if the rain gauge was not located in the center of the storm/event. Hence, a reduction by e.g. areal reduction factors is not included, since i) this would introduce another uncertainty and ii) is only common for load-case applications with using only extreme values as input. Also, the areal rainfall of the shown subcatchment results from 3 hourly time series and hence can be seen as representative for the area.

Nevertheless, we have implemented a discussion on that in the HBV-model description (p13 23-27).

[19 4] "flood quantiles are shown for a return period of 100 years" – It doesn't make sense as the observed period is much shorter. I would focus on 50 y return period to reduce the uncertainties.

The reviewer's suggestion is right, extrapolation is limited to the length of observations. We mention this one sentence later: "However, the extrapolation is limited by the length of the simulated runoff time series." (p24 2-4) The limitations are three times the runoff time series length and are indicated for each catchment by a dashed line in all figures showing extreme values. Results for higher return periods are not discussed in the manuscript. Nevertheless, for all three catchments extreme values are shown up to return periods of 100 years to enable comparisons between the figures (and for the catchment Pionierbrücke the extrapolation leads to return periods of 75 years.

[Fig. 12] For which catchment? Or is it for the entire region? Same comments as of Fig. 5 above.

The reviewer points out a missing information. The seasonal characteristics are estimated using all available stations for the entire regions with long time series (please see also the reply to the former comment). We have added the missing information in the manuscript in the same sentence (p28 3-5).

[25 5] to [25 10] More suitable to be in the Introduction section.

The sentences without results have been moved to the introduction.

[27 13] "It can be summarized, that the number of rain gauges has only a minor, but no systematic influence on runoff statistics for the catchments used in this investigation" – but likely not because of the number of rain-gauges in a catchment but because of the hydrological model that was used! I would like to see the same analysis using a fully distributed hydrological model that can account for spatial rainfall variability at the sub-catchment scales.

We see the reason for the reviewer's concern. We have discussed this in the model description (p13 14-20). For Pionierbrücke, additional simulations with WaSiM (150 m x 150 m spatial resolution) have also not shown systematic differences between the rainfall products (only slight differences for seasonal extreme values). Since Pionierbrücke was the catchments with our highest expectations to see differences resulting from the different rainfall products, we did not carry out further investigations for Tetendorf and Reckershausen.

[27 14] A repetitive.

We thank the reviewer for the hint and have deleted the sentence.

[Table 12 and 13] SI.

We thank the reviewer for the suggestion and have moved both tables to the SI.

[31 8] to [31 15] There are some repetitive here is well.

Thanks for the hint, we have removed the repetitive.

[32 9] "the IDW algorithm with an altitudinal rainfall adjustment, which was carried out by a linear regression model" – The IDW is likely to smooth the rainfall in space, thus reducing the spatial rainfall variability and the variability in flow.

We agree with reviewer 1, IDW leads to a smoothing of rainfall. However, the overall idea behind this investigation is to analyze the (non-, semi-) simultaneous occurrence of rainfall at different stations, which is assumed to cause the variability in space. A smoothing by IDW occurs for all rainfall products V1, V2 and V3 and the effect of the interpolation method should be minor in comparison to the effect of the spatial consistence.

[32 10] But FDC compare daily discharges, right? I guess that at hourly scale the differences are clearer.

The reviewer is concerned about the temporal resolution of the runoff time series used for the FDC. For the majority of the simulated periods, differences in the discharge during a day are small on an hourly time step. Hence, the daily resolution delivers satisfying results for FDC as a runoff statistic applied to represent the overall runoff behavior. For days with e.g. intense rainfall events and hence a high-dynamical runoff response the daily values are not representative. The representation of these instantaneous runoff values/peaks is very important and hence, these are represented by two of the four applied runoff statistics (summer and winter extremes) (please see also comment [12 10]). The authors do not think that a FDC based on hourly values would improve the calibration of the models and/or show greater differences between the different rainfall products. Also, the calibration and analyzes procedure based on a daily basis was applied before (e.g. Wallner and Haberlandt, 2015).

[38 6] to [38 11] Part of the reason is because the sub-catchments sets in SWMM model are often much finer than the 20-km set by HBV in this study. When exploring the hydrology response using small sub-catchments with SWMM the effect of the distributed rainfall in space are evident on the hydrological flows (see for example the study by Peleg et al. 2017, HESS). I have reasons to believe that if HBV model was set to have many more small sub-catchments for this study, the results of the differences between V1, V2 and V3 would look different. That rise the question of the suitability of HBV model with the current setting of a few large sub-catchment to explore the sensitivity of the hydrological response to different rainfall spatial characteristics.

We agree with the reviewer regarding the required spatial resolution for urban hydrology. Due to less storage and retention capabilities (no soil or vegetation the catchment reacts much faster. Hence, finer temporal and spatial resolution of rainfall is required. For catchments with a size of few hundred square kilometers, hourly resolution (Melsen et al., 2015) and thus a coarser spatial resolution is sufficient. The reviewer assumes higher differences between the rainfall products if the subcatchments would be smaller. However, this assumption was investigated by the application of WaSiM for catchment Pionierbrücke. Please see our reply to the comment [27 13].

[Conclusion section] The conclusion part is a mix of discussion, summary and conclusions. Consider revising it to make the outcome of the experiment clearer to the readers.

Indeed, the final section should also include a summary. Hence it was renamed to "Summary, conclusions and outlook". We follow the suggestion of the reviewer and have moved parts with discussions to the last but one section of the manuscript. However, we think the formulated open questions should remain in this section.

References:

Haberlandt, U., Eschenbach, A.-D. Ebner von und Buchwald, I. (2008). „A spacetime hybrid hourly rainfall model for derived flood frequency analysis". *Hydrology and Earth System Science* 12 (6), p. 1353–1367.

Herzog, Y. (2013): "Sensitivität eines hydrologischen Modells auf Veränderungen in den Klimavariablen Niederschlag, Temperatur und potentielle Verdunstung". *Student thesis* at the Institute of Water Resources Management, Hydrology and Agricultural Hydraulic Engineering, Faculty of Civil Engineering and Geodesy, Leibniz Universität Hannover (in German).

Melsen, L. A., Teuling, A. J., Torfs, P. J. J. F., Uijlenhoet, R., Mizukami, N. und Clark, M. P.: „Hydrology and Earth System Science Opinions: The need for process-based evaluation of large-domain hyper-resolution models". *Hydrology and Earth System Science* 20, 1069–1079, 2015.

Müller, H. und Haberlandt, U. (2015). „Temporal Rainfall Disaggregation with a Cascade Model: From Single-Station Disaggregation to Spatial Rainfall". *Journal of Hydrologic Engineering* 20 (11), p. 04015026.

Müller, H. und Haberlandt, U. (2018). „Temporal rainfall disaggregation using a multiplicative cascade model for spatial application in urban hydrology". *Journal of Hydrology 556*, p. 847-864.

Peleg, N., Marra, F., Fatichi, S., Paschalis, A., Molnar, P., Burlando, P.: Spatial variability of extreme rainfall at radar subpixel scale. J. Hydrol.

Wallner, M. and Haberlandt, U.: Klimabedingte Änderung von Hochwasserabflüssen im Aller-Leine-Einzugsgebiet - Eine Fallstudie mit HBV-IWW. Hydrol. Wasserbewirtsch. 59 (4), 174–183, 2015.

Wilks, D. S. (1998). „Multisite generalization of a daily stochastic precipitation generation model". *Journal of Hydrology* 210 (1-4), p. 178–191.

---

## Author Comment (AC4) · 5 Mar 2018

The manuscript without tracked changes can be found in the supplement.

Please also note the supplement to this comment:
https://www.hydrol-earth-syst-sci-discuss.net/hess-2017-609/hess-2017-609-AC4-supplement.pdf
* * *

---

## Author Comment (AC5) · 5 Mar 2018

**Rainfall disaggregation for hydrological modeling: Is there a need for spatial consistence?**

Hannes Müller, Markus Wallner, Kristian Förster

**- Supplementary material -**

[Figure]

**S1. Flow chart and applied calibration procedure for HBV (Wallner and Haberlandt, 2015)**

**S2. HBV model parameters modified during calibration with limiting ranges**

| Parameter | Unit | Explanation | Minimum | Maximum |
|---|---|---|---|---|
| *wsmf* | [mm$^{-1}$] | Wet snow melt factor | 1 | 4 |
| *tt* | [°C] | Threshold temperature | -1.5 | 1.5 |
| *dd* | [mm°C$^{-1}$d$^{-1}$] | Degree day factor | 0.5 | 5 |
| *fc* | [mm] | Field capacity | 50 | 300 |
| *lp* | [-] | Limit for potential evapotranspiration | 0.1 | 0.95 |
| *ß* | [-] | Empirical factor for runoff calculation from the soil layer | 0.5 | 4 |
| *hl* | [mm] | Threshold value for surface runoff | 1 | 30 |
| *k0* | [d] | Storage coefficient surface runoff | 0.25 | 5 |
| *k1* | [d] | Storage coefficient interflow | 3 | 40 |
| *k2* | [d] | Storage coefficient baseflow | 50 | 500 |
| *kperc* | [d] | Storage coefficient perculation | 3 | 40 |
| *maxbas* | [h] | Length of the triangular unit hydrograph impulse | 3 | 10 |
| *mx* | [-] | Weighting factor of Muskingum method | 0.1 | 0.4 |
| *mk* | [h] | Retention constant of Muskingum method | 0.25 | 10 |

[Figure]

**S3. Bivariate spatial rainfall characteristics of V1, V2 and V3 in comparison to observations for the catchment Reckershausen (for one realization, black circles represent observations - for details the reader is referred to Müller and Haberlandt (2015)).**

[Figure]

**S4. Bivariate spatial rainfall characteristics of V1, V2 and V3 in comparison to observations for the catchment Tetendorf (for one realization, black circles represent observations - for details the reader is referred to Müller and Haberlandt (2015)).**

[Figure]

**S5. Non-exceedance curve of areal rainfall intensities for V1, V2 and V3 for one subcatchment of Pionierbrücke (for one realization)**

**S6. NSC-values for all catchments and all criteria without calibration for validation period**

| Catchment | Criteria | V1 | V2 | V3 |
|---|---|---|---|---|
| Reckershausen | *Extr-Su* | 0.20 | 0.26 | 0.14 |
| | *Extr-Wi* | 0.76 | 0.77 | 0.77 |
| | *FDC* | 0.97 | 0.97 | 0.97 |
| | *Q-mon* | 0.99 | 0.99 | 0.99 |
| Pionierbrücke | *Extr-Su* | -1.68 | -1.58 | -1.59 |
| | *Extr-Wi* | 0.01 | 0.10 | 0.06 |
| | *FDC* | -0.07 | -0.07 | -0.07 |
| | *Q-mon* | 0.96 | 0.96 | 0.96 |
| Tetendorf | *Extr-Su* | 0.52 | 0.54 | 0.55 |
| | *Extr-Wi* | -7.78 | -7.78 | -8.41 |
| | *FDC* | 0.19 | 0.19 | 0.19 |
| | *Q-mon* | -0.06 | -0.05 | -0.11 |

**S7. $O_{stat}$-values for all catchments and all criteria without parameter calibration for validation period**

| Catchment | V1 | V2 | V3 |
|---|---|---|---|
| Reckershausen | 0.27 | 0.25 | 0.28 |
| Pionierbrücke | 1.14 | 1.10 | 1.11 |
| Tetendorf | 2.79 | 2.79 | 2.96 |

---

## Author Response (AR2)

**Editor comments:**

Dear authors, thank your for revisions, which have been eliminating most of the concerns by the reviewers. I think the concerns by reviewer 2 are valid and the authors need to be clearer on the use of daily time steps. in particular given the way the abstract and conclusions are written.

We thank the editor for pointing out this concern of reviewer 2. We have discussed it (see reply to concern #1 of reviewer 2) and implemented a brief explanation in the manuscript.

I do not understand the concept of neutral parameters. Arithmetic means of physical based parameters (even if they are just effective) is non physical. The authors need to explain their approach better.

We understand the concerns of the editor and the comment of reviewer on this issue. Please see our discussion in the reply to concern #2 of reviewer 2. Additionally, to the aforementioned reply, we want to add that although the parameters are physical interpretable, the assignment of one parameter value to each subcatchment represents already an averaging and is hence to a certain degree unphysical.

**Report #1 - Nadav Peleg**

The authors reply all comments from previous round. I do not have any major critics for the present version of the paper, but made some suggestions and comments for the authors to consider before accepting the paper for publication.

1. Equation #2 - "x" in the denominator should be deleted.
Thanks for the hint, we have replaced the multiplication sign by a different one.

2. p13 l28 - "smaller" should be "larger"?
No, "*smaller*" is correct for a temporal resolution of 1 hour. We have rephrased the sentence to clarify it. However, it has been analyzed for smaller distances, but for 5 min time steps.

3. P13 l29-31. This sentence is not clear to me.
This sentence represents a summary of the results for V2. However, due to the sentence before this was not clear. We have changed the sentence for clarification as follows:
Original: "a major improvement can be identified moving all station pairs into the cloud of observations (except some of the continuity ratio)."
Rephrased: "a major improvement for all characteristics can be identified by the application of V2, moving all station pairs into the cloud of observations (except some of the continuity ratio)."

4. p14 l14. "V2" should be "V3".
Reviewer 1 is correct, thanks for pointing it out.

5. p14 l15. Fig.5 or Fig. 4?
Reviewer 1 is correct, thanks for pointing it out.

6. p15 l5. "annual rainfall extremes" should be "annual maxima rainfall extremes".
We thank reviewer 1 for this hint, we have rephrased it.

7. p15 l15. You have only 1 realization - it can be that both product have the same statistics but from this realization it seems that V3 is higher than V2 for RP>18 years.
We are really thankful for this hint. After analyzing all 10 realizations applied for rainfall-runoff modeling, indeed the median of V2 and V3 are very similar. However, the conclusions are still valid

for the range of all realizations. We have rephrased the paragraph (p15) and replaced Fig. 5 with a new figure, including median and range of V1, V2 and V3.

8. I would change "investigation" to "study". For example, in p33 l3 instead of "The aim of this investigation is to investigate" will be something like "The aim of this study is to explore".
We thank reviewer 1 for this hint and have rephrased it in the manuscript several times.

9. Section 6 is a bit overloaded for a final section. Consider fragmenting to outlook and conclusion (making 2 sections out of this section).
We agree with reviewer 1 and have divided Section 6 into "Section 6 Summary" and "Section 7 Conclusions & Outlook".

**Report #2 - Anna Sikorska**

Specific comments:
1. My major concern is still related to the hypothesis tested by the authors i.e. the value of using "disaggregated rainfall products with different degrees of spatial consistence on rainfall-runoff modelling results" (abstract, first sentence). Particularly, the fact that the authors assess this effect only on cumulative statistics of runoff computed on a daily basis such as flow duration curves and monthly discharges rises concerns about the suitability of this type of analysis for the hypothesis tested in this paper. The explanation given by the authors in the reply to first reviews is not sufficient. In my opinion, the differences between different rainfall products may not be seen by the hydrological model if such cumulative statistics will be tested on a daily basis only. Keeping in mind the fact that the disaggregation of rainfalls is performed from daily rainfall values, it is not surprising that on a daily scale only minor differences (if at all) between different disaggregation schemes are observed. To judge whether the differences are observable or not, other metrics or similar metrics but at an hourly scale should also be investigated. Thus, sensitivity tests performed on a daily scale most likely cannot answer the question raised in the title. Moreover, if the aim of the rainfall disaggregation is to provide a high resolution data why would you assess the model performance on a daily and not on an hourly scale? And if the daily scale is of interest, why would you disaggregate the data at all? I think these issues should be clarified in the final version of the manuscript. If the authors insist on testing their hypothesis only on the cumulative values of daily runoff, this issue should also be more highlighted in the abstract and introduction, or even in the title.
We would like to thank reviewer 2 for her review and the critical remarks. We believe that the points addressed in your review are very helpful in the process of revising the manuscript. The main concern of reviewer 2 is the validation of the investigated methods with daily runoff statistics. We agree that a validation with only daily statistics would not be useful for the actual study. We had a short description on it in the submitted manuscript (p11 l9-11), but this brief part may not be sufficient for the reader. Hence, we have tried to emphasize that the focus of the actual study is set on the seasonal extreme values, Extr-Su and Extr-Wi. Both are analyzed with hourly time steps. A good representation of these two criteria is the motivation for the paper, since peak flows would be underestimated by simulations with daily time steps (Ding et al., 2016). We have tried to clarify this by rephrasing elements in the abstract (p1 l21-24) and in the manuscript itself (p16 l11-13, p31 l2-5). The runoff statistics Q-mon and FDC as cumulative runoff statistics are additionally applied to train and validate the model not only on extreme events what might have lead to implausible parameter sets which are not representing the general behavior of the catchments.

2. The use of a calibrated or a non-calibrated hydrological model needs a further explanation. Particularly, what do the authors mean with "the neutral parameter set"? This is not a common hydrological term. As it is written in the discussion, "the neutral parameter set" was derived from parameter ranges by simply using an arithmetic mean. If you use an arithmetic mean, no correlations between parameters will be preserved which may lead to an unrealistic model behaviour. Did you check how does the model behave with using arithmetic means for model parameters? Moreover, these means will strongly depend on the ranges chosen for model parameters (How these ranges were exactly defined?) A better choice would be to use default (or regionalized) values for model parameters which are derived from analysis with a large sample of catchments (at least for the HBV model which is commonly applied in the hydrological community) or use a calibrated model on independent time series (observed hourly data if available).

We agree with the reviewer that the explanations on the 'neutral' parameter set could be improved. the term "neutral parameter set" is not common, which is why we revised the corresponding paragraph in our manuscript: "For each parameter, the arithmetic mean of the upper and lower bound for each parameter (as described by Wallner et al., 2013, see also Supplementary material S2) is utilized to form what is called a 'neutral' parameter set here. In this context, the term 'neutral' means that the parameter set is seen as a default parameter set for the study region. Moreover, this approach provides feasible and hydrological meaningful estimates of the parameters without favoring V1, V2 or V3. Thus, the parameters are chosen independently from the rainfall products making them suitable for comparing different rainfall inputs." We agree with reviewer 2 that the application of calibrated parameters using independent time series would be better, but as discussed in the manuscript on p26, the data is not available. Indeed, the application of this neutral parameter set is a pragmatic approach and has several shortcomings. It is beyond the scope of this study to discuss possible deficiencies regarding lower correlations among parameters. However, using an average value out of the range of possible values provided for a certain parameter is a common way to iterate model parameters in optimization algorithms. Using a value in-between a suggested minimum and maximum value might be seen as a default value. Since the values provided by Wallner et al. (2013) are regional values found for catchments in Lower Saxony, a regional relation is considered. We agree that averaging parameters is not feasible in general. However, using a value in-between typical ranges is considered as a first guess and a starting point from where to start. We do not attempt to introduce this pragmatic approach as a common approach to estimate parameters. Our intention was to provide a parameter set that is independent from calibrations with observed rainfall data, which in turn might has stronger similarities to a certain rainfall product and hence might introduces biases in our comparison. The results found for simulations utilizing the neutral parameter set are indeed less accurate but they provide reliable estimates of the hydrological response of the catchments. We added some more explanations in the manuscript. Thanks for your comment. We believe that this really helped to improve the manuscript.

3. Regarding the units of model parameters, use of days in the table with parameter ranges is still misleading. Please specify under the table with model parameters which parameters are defined in hours and which in days.
We assume this comment refers mainly to the storage coefficients for surface runoff, interflow and baseflow. From the range of the parameter it is visible, that for e.g. the surface runoff coefficient the lower limit is 6 h (0.25 d). So although the unit shown is [d], also values below 1 d are possible.

However, we have chosen the unit [d] for all three coefficients to enable comparisons of the ranges for the three coefficients. Hence, we prefer to not change the units in the table, which was taken from Wallner et al. (2013).

[revised manuscript text omitted]

---

## Author Response (AR3)

Editor Decision: Publish subject to minor revisions (review by editor) (16 Aug 2018) by Florian Pappenberger

We are thankful for the additional review by Florian Pappenberger and his useful comments. We have involved our replies in the manuscript.

Comments to the Author:

Comment 1) use of daily and hourly metrics:

Indeed, focusing on seasonal extreme values (Extr-Su and Extr-Wi) analysed at an hourly time step seems to be a more reasonable approach than analysing only summarized runoff statistics based on daily data. I am still surprised that not much differences are seen but this may require a more detailed analysis exceeding the scope of this paper.

We are thankful for this comment and we have also expected more differences between the different rainfall products. A more detailed analysis is indeed required, but as the editor mentions, this would go beyond the scope of the actual study. We are working on this topic in a new research project. However, we think a communication of the actual results, which are surprising, is useful for the community.

Comment 2) "neutral parameter sets"

I am still not convinced about introducing this term for an arithmetic mean of defined parameter ranges at regional scale. Despite its shortcomings which were not discussed in the manuscript neither in the authors' reply, in my opinion, use of a common definition "default parameter set" would be more appropriate, particularly because that is what authors mean by their 'neutral' parameter set. The authors also state in their response that they do not intended to propose this method for parameter estimation: "We do not attempt to introduce this pragmatic approach as a common approach to estimate parameters…". This is why they should stick to common terms already used in hydrology instead of introducing new (potentially misleading) terms which after publication may be used by other researchers. An alternative choice to a default parameter set would be an independent parameter set.

We agree with the editor that the introduction of the term ‚neutral parameter set' could lead to confusion in the community, hence we changed it as suggested to ‚default parameter set'. Regarding the shortcomings of this approach, we implemented the following sentence: „The application of a default parameter set includes some shortcomings, e.g. regarding the physical interpretability, but it enables a comparison of the rainfall products.". (p26 l17-18)

Comment 3) units in days/hours.

The answer is ok, however the major confusion arrives from the fact that the parameters are defined in days whereas the model is run at hourly time step. A single sentence below the table clarifying this issue would solve the problem

We are thankful for the idea of the editor and added the following information to the table caption for clarification: „(please note that although for some parameters the values are given in days, all models are run at hourly time step.)"

[revised manuscript text omitted]